# Thermophysical Model for Online Optimization and Control of the Electric Arc Furnace

**Sudi Jawahery** [1,*] **, Ville-Valtteri Visuri** [2,3] **, Stein O. Wasbø** [1] **, Andreas Hammervold** [1] **, Niko Hyttinen** [2] **and Martin Schlautmann** [4]

1   Cybernetica AS, Leirfossvegen 27, 7038 Trondheim, Norway; stein.wasbo@cybernetica.no (S.O.W.); andreas.hammervold@cybernetica.no (A.H.)
2   Research and Development, Outokumpu Stainless Oy, Terästie, 95490 Tornio, Finland; ville-valtteri.visuri@outokumpu.com (V.-V.V.); niko.hyttinen@outokumpu.com (N.H.)
3   Process Metallurgy Research Unit, University of Oulu, 90014 Oulu, Finland
4   VDEh-Betriebsforschungsinstitut GmbH, Sohnstraße 69, 40237 Düsseldorf, Germany; Martin.Schlautmann@BFI.de
*   Correspondence: sudi.jawahery@cybernetica.no

**Abstract:** A dynamic, first-principles process model for a steelmaking electric arc furnace has been developed. The model is an integrated part of an application designed for optimization during operation of the furnace. Special care has been taken to ensure that the non-linear model is robust and accurate enough for real-time optimization. The model is formulated in terms of state variables and ordinary differential equations and is adapted to process data using recursive parameter estimation. Compared to other models available in the literature, a focus of this model is to integrate auxiliary process data in order to best predict energy efficiency and heat transfer limitations in the furnace. Model predictions are in reasonable agreement with steel temperature and weight measurements. Simulations indicate that industrial deployment of Model Predictive Control applications derived from this process model can result in electrical energy consumption savings of 1–2%.

**Keywords:** electric arc furnace; mathematical modeling; model predictive control

## 1. Introduction

Electrical arc furnaces (EAF) perform a primary steelmaking process that converts recycled steel scrap into liquid steel, which can be refined further in downstream processes. The EAF is a refractory-lined vessel that is filled with steel scrap at the start of each new heat. Through holes in the vessel roof, graphite electrodes (a single electrode in DC furnaces and and three electrodes in AC furnaces) are lowered and used to conduct a high-voltage electric arc that supplies electrical energy to melt the scrap metal. Gas burners are mounted along the outer vessel sidewalls. During the course of a heat, the burners can operate in two different modes: (1) by providing pure oxygen for refining, or (2) by providing a mixture of oxygen and either liquefied natural gas (LNG) or propane to burn for extra heating. The burners are typically operated in fuel combustion mode during the early process stages, while refining takes place towards the end of the heat. The use of gas burners has been shown to decrease batch time and reduce electrical power consumption. To protect the vessel and furnace equipment from sustaining damage due to radiation from the electric arc and heated metal, cooling water heat exchange panels are mounted along the upper parts of the vessel's sidewalls and the roof [1].

A heat is typically run as either a one-, two- or three-basket heat. This means the vessel is charged with scrap metal one, two or three times during a heat, with the first basket always being charged before the electric arc and gas burners are turned on. The baskets can vary significantly, both in size and in the type of scrap being charged. Carbon and additional slag-forming materials are also added to the furnace in order to achieve the

desired slag-phase composition and foaming. At the end of each heat, the slag phase and liquid steel are tapped separately from the furnace, and a new heat is ready to begin [1].

Numerous mathematical models have been proposed for predicting the course of the EAF process. Recently, Hay et al. [2] presented a comprehensive review of mathematical models proposed to date. They concluded that while there are still several development areas, modern models can predict the main dynamic changes in distribution of species and energy with reasonable accuracy. Furthermore, it was suggested that fundamental models are now sufficiently fast to be used for model predictive control (MPC).

Some relevant studies [3–7] on the application of MPC for the EAF process are summarized in the following sentences. The model by Bekker et al. [3] is intended for controlling the offgas system and manipulates two variables (fan force and slip-gap) to adjust three outputs: the relative furnace pressure, offgas temperature and offgas CO mass fraction. Of these three variables, the relative furnace pressure was regulated, while the offgas temperature and offgas CO mass fraction were only limited. Extending the model by Bekker et al. [3], Oosthuizen et al. [4] presented a slag foaming model and introduced the rate of direct reduced iron (DRI) addition as an additional manipulated input variable. Later, Oosthuizen et al. [5] proposed a related MPC algorithm based on economic objectives. The MPC proposed by MacRosty and Swartz is formulated in terms of an economic performance objective. More specifically, the model adjusts the arc power, oxygen flow from the burner, natural gas flow from the burner, oxygen injection, carbon injection, and mass of the second charge to minimize the total costs of the EAF process. Shyamal [7] proposed a shrinking horizon MPC algorithm, which was coupled with multi-rate moving horizon estimation (MHE) for real-time model calibration. The model was directed at real-time energy management and employed time-varying electricity prices for decision making. Shyamal [7] also proposed a real-time dynamic advisory system, which was based on multi-tiered optimization of the estimated states from MHE. It is worth noting that the MPC algorithm employed by Oosthuizen et al. [4,5] is linear, while those employed by Bekker et al. [3], MacRosty and Swartz [6] and Shyamal [7] are non-linear.

A model comprising monitoring and prediction of thermal and metallurgical heat state evolution in the EAF has been developed by BFI [8–10]. This dynamic EAF process model uses event driven and cyclically measured process data to calculate the temperature, weights and analyses of the steel and slag phases in the furnace. The model considers these phases without spatial resolution and uses ordinary differential equations in time and algebraic equations to describe the process state. The same model kernel can be used to monitor the current heat state from actual process data and to predict its further evolution based on related practice data for the remaining treatment steps. In order to monitor the thermal process state, the BFI model calculates the current energy content of the melt based on a cyclically evaluated overall energy balance. The energy into the balance is the sum of the electrical energy supplied and the chemical energy released by reactions. The energy leaving the balance takes into account the losses to cooling water, offgas, radiation and convection. The bath temperature is obtained from the difference of the current energy content and the energy requirement for meltdown, which is in turn calculated from the reference enthalpies (i.e., specific enthalpies at reference temperature) of the charged materials (scrap and slag formers), where the hot heel is also taken into account.

The monitoring of the metallurgical process state in the BFI model comprises the cyclic calculation of the weight and the composition for the metal bath and the slag phase. For this purpose, the input by the charged materials as well the effects of the different oxidation and reduction reactions (decarburization, dephosphorization and slagging/reduction of metallic elements) are considered. The latter are based on appropriate first-order differential equations where the reaction rate of an element or oxide is given by its content in steel or slag multiplied by the oxygen or reduction agent input rate and an adapted oxidation or reduction efficiency, respectively.

The aim of this work was to formulate a new dynamic model of an EAF that could be used to optimize the electric power profile and electric arc operation. For that purpose, it

goes a step beyond the mentioned state-of-the art approaches and uses a more detailed modelling of different control volumes with liquid and solid phases. The model in this work is non-linear, allowing for the representation of complex interrelated phenomena, including estimation of the visibility of the electric arc and arc efficiency for melting and heating. The MPC application of the proposed model uses a finite receding horizon, where the duration of the heat is not specified, but the predictions horizon becomes shorter as the process nears the completion criteria. To validate the viability of the new approach, the model and related MPC were tested for an industrial scale EAF in stainless steel-making. While Visuri et al. [11] presented some preliminary results, extensive results along with accompanying interpretation and explanation are discussed in this article.

## 2. Materials and Methods

The process model developed for the EAF is based on physical modeling principles such as mass and energy balances. When creating a process model, it is essential that the model can provide the necessary information for solving the problem at hand without becoming computationally cumbersome. In this case, the model is designed to be used as a basis for real-time optimization. Hence, the focus of the modeling efforts is to ensure that the model is fast enough to be recalculated multiple times in each time sample. The model is developed as a continuous-time model that is integrated over selected time steps using the forward Euler method for numerical integration. The numerical smoothness of the model has been emphasized for two reasons: (1) so that computationally fast explicit integration schemes can be applied without losing accuracy, and (2) so that optimization problems formulated with model output can be more easily designed to be convex [12]. Further, a Kalman filter (KF) has been designed to ensure that the process model follows the efficiency of the real process. Measurement outputs are calculated by the model and compared with process measurements. The residuals between the model predictions and real measurements are fed into the KF, which updates state variables and selected parameters for estimation [13].

### 2.1. Control Volumes and State-Space Variables

Figure 1 shows the process state variables that are included in the model. Energy supplied by three electrodes is used to directly heat the contents of the inner solid and liquid masses. As a result, the inner control volume has the highest temperatures in the model and is therefore colored in red, with bordering masses (the outer control volume, gas) colored in orange gradients. The temperatures, total masses and masses of individual components in the solid and liquid phases of the inner and outer steel control volumes are modeled. The temperature, total masses and individual component masses of the solid and liquid slag phases are also modeled. The component masses are enumerated in Table 1. The environment is modeled by including dynamic states for the temperatures of the roof, side panels and gas that fills the space not occupied by steel in the furnace. The roof and side panels are both in contact with cooling water streams for which the temperature measurement is recorded. Calculation of the cooling water temperature variation as predicted by the model allows for real-time comparison to process measurements. The temperature of the process offgas is recorded downstream in a duct that extracts fumes from the furnace. The offgas temperature in the duct is modeled accordingly and also compared to real-time data.

In order to maintain the model's focus on the energy balance, the modeled slag masses exchange heat and mass only with the steel and not with the furnace environment. This assumption reduces the model complexity and allows the parameter estimation discussed in Section 2.5 to more directly impact the states of interest, namely the solid and liquid steel. Heat transfer between steel and the slag masses is then tuned to indirectly account for the interactions of slag with the environment. Slag properties are taken from Jiao et al. [14], and properties for the furnace materials are taken from Fruehan [1].

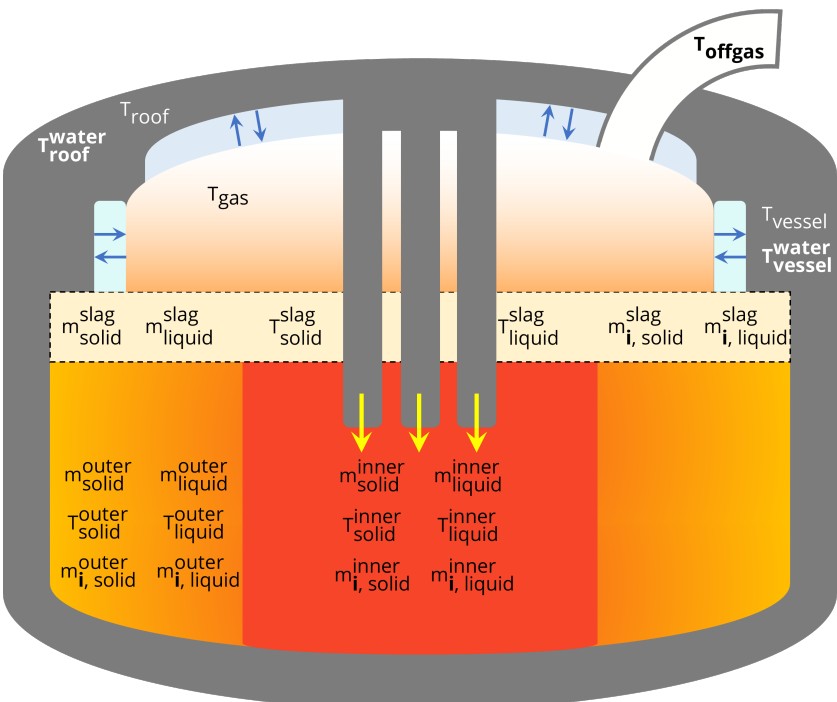

**Figure 1.** State variables in the EAF process model. $T_{roof}^{water}$, $T_{vessel}^{water}$ and $T_{offgas}$ can be compared to real-time process data.

**Table 1.** Modeled components in the steel and slag phases.

| Dissolved Component | Phase(s) | Reactive with $O_2$ in Model? | Equilibrium Reaction in Model? |
|---|---|---|---|
| Fe | Liquid, Solid | Yes | Yes |
| C | Liquid, Solid | Yes | Yes |
| Cr | Liquid, Solid | Yes | Yes |
| Si | Liquid, Solid | Yes | Yes |
| Al | Liquid, Solid | Yes | Yes |
| Mn | Liquid, Solid | No | Yes |
| FeO | Slag | No | Yes |
| $SiO_2$ | Slag | No | Yes |
| $Cr_2O_3$ | Slag | No | Yes |
| $Al_2O_3$ | Slag | No | Yes |
| MnO | Slag | No | Yes |

Because the focus of this model is to achieve a dynamic energy balance rather than a detailed mass balance, not all components recorded in process data are included in the model. The components listed in Table 1 represent the components whose non-oxide element represent more than 2% of charged mass and whose reactions have the potential to significantly affect the energy balance.

### 2.2. Chemical Reactions

Gas burners supply flows of LNG and oxygen that react to release energy:

$$\{CH_4\} + 2\{O_2\} \rightarrow 2\{H_2O\} + \{CO_2\}. \tag{1}$$

Oxygen that is not fully consumed by the reaction with LNG, for example during the refining phase, can react with CO gas and components in the liquid steel to form liquid slag components and gases:

$$\{CO\} + \frac{1}{2}\{O_2\} \rightarrow \{CO_2\}$$
$$Fe + \frac{1}{2}\{O_2\} \rightarrow (FeO)$$
$$[C] + \frac{1}{2}\{O_2\} \rightarrow \{CO\}$$
$$[C] + \{O_2\} \rightarrow \{CO_2\} \qquad (2)$$
$$[Si] + \{O_2\} \rightarrow (SiO_2)$$
$$2[Cr] + \frac{3}{2}\{O_2\} \rightarrow (Cr_2O_3)$$
$$2[Al] + \frac{3}{2}\{O_2\} \rightarrow (Al_2O_3).$$

The oxygen for the reactions in Equation (Equation (2)) is divided between the CO gas and the liquid steel components. Oxygen is allocated to the inner and outer liquid steel proportionally based on their masses. Within each control volume, the oxygen consumption in Equation (Equation (2)) is proportional to the mass fractions of Fe, C, Si, Cr and Al.

The model's explicit treatment of oxygen consumption by reactions in (Equation (2)) does not account for the activity coefficients of the different components. However, thermodynamic equilibrium is enforced by the inclusion of reversible reactions between the liquid steel and slag phases. Including these equilibrium reactions in the model achieves component mass fraction ratios that are consistent with the equilibrium constants given in Appendix A.2, which have been adapted from Turkdogan [15]. The following steel–slag equilibrium reactions take place, where both the forward and backward reactions are modeled:

$$(FeO) + [C] \leftrightarrow Fe + \{CO\}$$
$$(FeO) + [Mn] \leftrightarrow Fe + (MnO)$$
$$(MnO) + [C] \leftrightarrow [Mn] + \{CO\}$$
$$2(FeO) + [Si] \leftrightarrow 2Fe + (SiO_2) \qquad (3)$$
$$2(MnO) + [Si] \leftrightarrow 2[Mn] + (SiO_2)$$
$$3(FeO) + 2[Cr] \leftrightarrow 3Fe + (Cr_2O_3)$$
$$3(SiO_2) + 4[Al] \leftrightarrow 3[Si] + 2(Al_2O_3).$$

Equations (1)–(3) form a pared down version of the reactions modeled by Logar et al. [16], with the exception of the reactions involving Al and $Al_2O_3$.

The energy released and consumed by the reactions in Equations (1)–(3) is distributed between different masses in the furnace model. Energy released by Equation (1) is split between the steel and the gas, where the efficiency of LNG burning to heat steel changes during the process as described by Logar et al. [17]. All of the energy released by Equations (2) and (3) heats the steel. The steel-heating reaction energy is first divided between the inner and outer steel proportionally based on their volumes, as defined by the dimensions given in Appendix A.4. The model distributes the total reaction energy to the entire furnace contents because distributing energy based on the control volume of the reactants produces physically unreasonable results. For example, using all of the reaction energy to heat the inner steel when only the inner steel contains liquid phase reactants leads to excessively high temperatures and unreasonable temperature gradients. Within each control volume, the reaction energy is divided between the solid and liquid phases proportionally based on mass.

*2.3. Heat Transfer*

Heat transfer in the model is simplified such that all energy supplied to and lost by the furnace contents is exchanged exclusively with the steel masses. The heating of other furnace contents, namely slag, is then accounted for indirectly by tuning the heat transfer between the steel and slag masses. Heat transfer between some masses are modeled linearly, while others are accounted for only by radiation.

Convection and conduction between select masses in the furnace are modeled as linear heat transfer:

$$Q_{ij} = k_{ij} A_{ij} (T_i - T_j). \tag{4}$$

In Equation (4), the amount of heat flowing from mass $i$ to mass $j$ is proportional to the temperature difference $(T_i - T_j)$, heat transfer coefficient $k_{ij}$ and area for heat transfer $A_{ij}$. A comprehensive list of all masses involved in linear heat transfer along with the corresponding coefficients and areas is available in Table 2.

**Table 2.** Prefactors for linear heat transfer between different masses in the furnace. The subscript letters (s, l, c, b, r, v, g) refer to (solid steel, liquid steel, solid slag, liquid slag, roof, vessel, gas), respectively. An empty table entry indicates that linear heat transfer between the two masses is omitted from the model.

| | Inner Solid | Outer Solid | Inner Liquid | Outer Liquid | Solid Slag | Liquid Slag | Roof | Vessel | Gas |
|---|---|---|---|---|---|---|---|---|---|
| **Inner Solid** | - | $k_{ss} A_{ss}^{cross}$ | $k_{sl} A_{sl}^{inner}$ | $k_{sl} A_{sl}^{cross}$ | $k_{cs} A_{cs}^{inner}$ | $k_{bs} A_{bs}^{inner}$ | - | - | $k_{sg} A_{bs}^{inner}$ |
| **Outer Solid** | $k_{ss} A_{ss}^{cross}$ | - | $k_{sl} A_{ls}^{cross}$ | $k_{sl} A_{sl}^{outer}$ | $k_{cs} A_{cs}^{outer}$ | $k_{bs} A_{bs}^{outer}$ | - | - | $k_{sg} A_{bs}^{outer}$ |
| **Inner Liquid** | $k_{sl} A_{sl}^{inner}$ | $k_{sl} A_{ls}^{cross}$ | - | $k_{ll} A_{ll}^{cross}$ | $k_{cl} A_{cl}^{inner}$ | $k_{bl} A_{bl}^{inner}$ | - | - | $k_{lg} A_{bl}^{inner}$ |
| **Outer Liquid** | $k_{sl} A_{sl}^{cross}$ | $k_{sl} A_{sl}^{outer}$ | $k_{ll} A_{ll}^{cross}$ | - | $k_{cl} A_{cl}^{outer}$ | $k_{bl} A_{bl}^{outer}$ | - | - | $k_{lg} A_{bl}^{outer}$ |
| **Solid Slag** | $k_{cs} A_{cs}^{inner}$ | $k_{cs} A_{cs}^{outer}$ | $k_{cl} A_{cl}^{inner}$ | $k_{cl} A_{cl}^{outer}$ | - | - | - | - | - |
| **Liquid Slag** | $k_{bs} A_{bs}^{inner}$ | $k_{bs} A_{bs}^{outer}$ | $k_{bl} A_{bl}^{inner}$ | $k_{bl} A_{bl}^{outer}$ | - | - | - | - | - |
| **Roof** | - | - | - | - | - | - | - | - | $k_{gr} A_r$ |
| **Vessel** | - | - | - | - | - | - | - | - | $k_{gv} A_v$ |
| **Gas** | $k_{sg} A_{bs}^{inner}$ | $k_{sg} A_{bs}^{outer}$ | $k_{lg} A_{bl}^{inner}$ | $k_{lg} A_{bl}^{outer}$ | - | - | $k_{gr} A_r$ | $k_{gv} A_v$ | - |

To model the area for heat transfer between solid and liquid phases within each steel control volume, the scrap metal is assumed to linearly transition from being a single slab at the beginning of the process (solid mass fraction $x_{solid} \approx 1$) to small solid particles at the end of the process (solid mass fraction $x_{solid} \approx 0$) in a manner that resembles the melting phenomenon described by González et al. [18]. This transition is modeled as:

$$r_{particle}^{max} = \left( \frac{3 m_{solid}}{4 \rho_{solid} \pi} \right)^{1/3}$$
$$r_{particle} = \left( r_{particle}^{max} - r_{particle}^{min} \right) x_{solid} + r_{particle}^{min}. \tag{5}$$

Equation (5) means that the maximum particle size in either control volume is a function of the instantaneous mass of solid $m_{solid}$ and the solid fraction, where $m_{solid}$ refers either to the model states $m_{solid}^{outer}$ or $m_{solid}^{inner}$ as depicted in Figure 1. The maximum particle radius is first calculated by assuming the whole mass of the solid to be a single spherical particle. This assumption is then corrected for by relating the actual particle radius size to the solid mass fraction. When $x_{solid} < 1$, the mass of solid is assumed to be broken up, leading to smaller particle radii. The radii of solid particles decrease until they reach the model constant $r_{particle}^{min} = 10 \, cm$, at which point the particle's mass is assumed to be purely virtual. The area for solid–liquid heat transfer is then modeled as a function of both the liquid mass fraction $x_{liquid}$, the solid mass and $r_{particle}$:

$$A_{sl} = \frac{3 x_{liquid} m_{solid}}{\rho_{solid} r_{particle}}. \tag{6}$$

The liquid fraction factor in Equation (6) accounts for the liquid coverage of the solid particles: at small $x_{\text{liquid}}$, the entire surface area of the solid particles may not be in contact with liquid metal. Equation (6) is applied directly to calculate $A_{\text{sl}}^{\text{inner}}$ and $A_{\text{sl}}^{\text{outer}}$. The area for heat transfer between unlike phases in different control volumes is also calculated from Equation (6), but the result is scaled by a factor of 3 to account for reduced mixing between the control volumes and the substitution of terms depends on the specific combination of phases being modeled ($A_{\text{sl}}^{\text{cross}} = \frac{x_{\text{liquid}}^{\text{outer}} m_{\text{solid}}^{\text{inner}}}{\rho_{\text{solid}} r_{\text{particle}}^{\text{inner}}}$; $A_{\text{ls}}^{\text{cross}} = \frac{x_{\text{liquid}}^{\text{inner}} m_{\text{solid}}^{\text{outer}}}{\rho_{\text{solid}} r_{\text{particle}}^{\text{outer}}}$).

To model the area for heat transfer between like phases in different control volumes, the total area separating the control volumes is first calculated and then scaled with the appropriate phase fractions:

$$
\begin{aligned}
h_{\text{inner}} &= \frac{\frac{m_{\text{solid}}^{\text{inner}}}{\rho_{\text{solid}}} + \frac{m_{\text{liquid}}^{\text{inner}}}{\rho_{\text{liquid}}}}{A_{\text{inner}}} \\[2ex]
h_{\text{outer}} &= \frac{\frac{m_{\text{solid}}^{\text{outer}}}{\rho_{\text{solid}}} + \frac{m_{\text{liquid}}^{\text{outer}}}{\rho_{\text{liquid}}}}{A_{\text{outer}}} \\[2ex]
A_{\text{cross}} &= \pi d_{\text{inner}} \frac{h_{\text{inner}} + h_{\text{outer}}}{2} \\[2ex]
A_{(\text{ss/ll})}^{\text{cross}} &= x_{(\text{solid/liquid})}^{\text{inner}} x_{(\text{solid/liquid})}^{\text{outer}} A_{\text{cross}}.
\end{aligned}
\tag{7}
$$

The inner and outer control volume areas and diameters are defined by the model dimensions given in Appendix A.4.

The areas for heat transfer between steel and solid slag are calculated based on the solid slag mass, the specific area of slag $A_{\text{s}}$ given by Bekker et al. [19] and the appropriate phase fractions:

$$
\begin{aligned}
A_{\text{c}}^{(\text{inner/outer})} &= m_{\text{c}} A_{\text{s}} \frac{A_{(\text{inner/outer})}}{A_{\text{inner}} + A_{\text{outer}}} \\[2ex]
A_{\text{c(s/l)}}^{(\text{inner/outer})} &= x_{(\text{solid/liquid})}^{(\text{inner/outer})} A_{\text{c}}^{(\text{inner/outer})}.
\end{aligned}
\tag{8}
$$

The areas for heat transfer between steel and liquid slag are calculated based on the metal bath surface area because the liquid slag forms as a layer on top of the steel. These areas are also applicable for the heat transfer between steel and the surrounding gas:

$$
A_{\text{b(s/l)}}^{(\text{inner/outer})} = x_{(\text{solid/liquid})}^{(\text{inner/outer})} A_{(\text{inner/outer})}.
\tag{9}
$$

Although the roof and side panel cooling water does not exchange heat directly with the steel, the linear heat flux from the furnace to the cooling water must be calculated in order to predict the outlet water temperature and compare to process data:

$$
Q_{(\text{r/v})\text{w}} = k_{(\text{r/v})\text{w}} A_{(\text{r/v})} \left( T_{(\text{r/v})} - T_{(\text{r/v})}^{\text{w}} \right).
\tag{10}
$$

Radiation between steel and the furnace surfaces is included in the model. Table 3 lists the equations for heat flux from the steel to the furnace roof and vessel $Q_{(\text{s/l})(\text{r/v})}^{(\text{inner/outer})}$.

**Table 3.** Radiative heat transfer between different masses in the furnace. The subscript letters (s, l, b, r, v) refer to (solid steel, liquid steel, liquid slag, roof, vessel), respectively.

|  | Roof | Vessel |
|---|---|---|
| **Inner Solid** | $\sigma_{\text{SB}} A_{\text{bs}}^{\text{inner}} VF_{\text{r}}^{\text{inner}}$ $\left( \varepsilon_s T_{\text{s}}^{\text{inner}^4} - \varepsilon_r T_{\text{r}}^4 \right)$ | $\sigma_{\text{SB}} A_{\text{bs}}^{\text{inner}} VF_{v}^{\text{inner}}$ $\left( \varepsilon_s T_{\text{s}}^{\text{inner}^4} - \varepsilon_r T_{\text{v}}^4 \right)$ |
| **Outer Solid** | $\sigma_{\text{SB}} A_{\text{bs}}^{\text{outer}} VF_{\text{r}}^{\text{outer}}$ $\left( \varepsilon_s T_{\text{s}}^{\text{outer}^4} - \varepsilon_r T_{\text{r}}^4 \right)$ | $\sigma_{\text{SB}} A_{\text{bs}}^{\text{outer}} VF_{\text{v}}^{\text{outer}}$ $\left( \varepsilon_s T_{\text{s}}^{\text{outer}^4} - \varepsilon_r T_{\text{v}}^4 \right)$ |
| **Inner Liquid** | $\sigma_{\text{SB}} A_{\text{bl}}^{\text{inner}} VF_{\text{r}}^{\text{inner}}$ $\left( \varepsilon_l T_{\text{l}}^{\text{inner}^4} - \varepsilon_r T_{\text{r}}^4 \right)$ | $\sigma_{\text{SB}} A_{\text{bl}}^{\text{inner}} VF_{\text{v}}^{\text{inner}}$ $\left( \varepsilon_l T_{\text{l}}^{\text{inner}^4} - \varepsilon_r T_{\text{v}}^4 \right)$ |
| **Outer Liquid** | $\sigma_{\text{SB}} A_{\text{bl}}^{\text{outer}} VF_{\text{r}}^{\text{outer}}$ $\left( \varepsilon_l T_{\text{l}}^{\text{outer}^4} - \varepsilon_r T_{\text{r}}^4 \right)$ | $\sigma_{\text{SB}} A_{\text{bl}}^{\text{outer}} VF_{\text{v}}^{\text{outer}}$ $\left( \varepsilon_l T_{\text{l}}^{\text{outer}^4} - \varepsilon_r T_{\text{v}}^4 \right)$ |

$\sigma_{\text{SB}}$ is the Stefan–Boltzmann constant. The view factors $VF_{\text{v}}^{\text{inner}}$ and $VF_{\text{v}}^{\text{outer}}$ are calculated based on equations for disks (inner steel) and annular rings (outer steel) embedded in the base of a cylinder to the cylinder column (vessel) [20]. Because no radiative heat transfer from steel to gas or between steel phases is included in the model, the roof is the only other surface that absorbs steel radiation, and the view factors between the steel control volumes and the roof are solved for ($VF_{\text{r}}^{\text{inner}} = 1 - VF_{\text{v}}^{\text{inner}}$; $VF_{\text{r}}^{\text{outer}} = 1 - VF_{\text{v}}^{\text{outer}}$).

Radiative heat transfer from the roof to the furnace vessel is given by:

$$Q_{\text{rv}} = \sigma_{\text{SB}} A_{\text{r}} VF_{\text{v}}^{\text{r}} \left( \varepsilon_r T_{\text{r}}^4 - \varepsilon_v T_{\text{v}}^4 \right). \tag{11}$$

The view factor $VF_{\text{v}}^{\text{r}}$ in Equation (11) is calculated based on an equation for the base of a cylinder (roof) to the cylinder column (vessel) [20].

### 2.4. Solid–Liquid Phase Change

Melting and freezing are a mass transfer mechanism between the solid and liquid phases in the inner steel, outer steel and slag. Melting is assumed to take place gradually as the solid temperature increases in a range centered around a defined melting/liquidus temperature $T_{\text{m}}$. Similarly, freezing takes place as the liquid temperature decreases in the same range. When both solid and liquid temperatures are within this range, melting and freezing take place simultaneously with temperature-dependent rates.

The melting and freezing mechanisms are illustrated in Figure 2. During the melting process, the solid temperature increases above the lower boundary for the phase change region. Liquid mass begins to accumulate and the liquid temperature changes quickly from its original virtual value. Eventually, the solid mass disappears, and additional energy inputs heats the liquid phase. Analogously, the liquid temperature decreases below the upper boundary for the phase change region to start the freezing process. Solid mass accumulates and the solid temperature changes quickly from its original virtual value. Eventually, the liquid mass disappears, and the solid mass continues to cool.

The rates of melting and freezing are given by:

$$\begin{aligned} r_{\text{melt}} &= k_{\text{phase}} m_{\text{s}} \frac{\max(0, (T_{\text{solid}} + dT_{\text{m}}) - T_{\text{m}})}{2dT_{\text{m}}} \\ r_{\text{freeze}} &= k_{\text{phase}} m_{\text{l}} \frac{\max\left(0, T_{\text{m}} - \left(T_{\text{liquid}} - dT_{\text{m}}\right)\right)}{2dT_{\text{m}}}. \end{aligned} \tag{12}$$

The heat of fusion determines the amount of energy exchanged between the two phases:

$$\begin{aligned} Q_{\text{melt}} &= \Delta H_{\text{fusion}} r_{\text{melt}} \\ Q_{\text{freeze}} &= -\Delta H_{\text{fusion}} r_{\text{freeze}}. \end{aligned} \tag{13}$$

The terms for the heat of melting and freezing of each phase contribute to the energy balance equations found in Equation (A21).

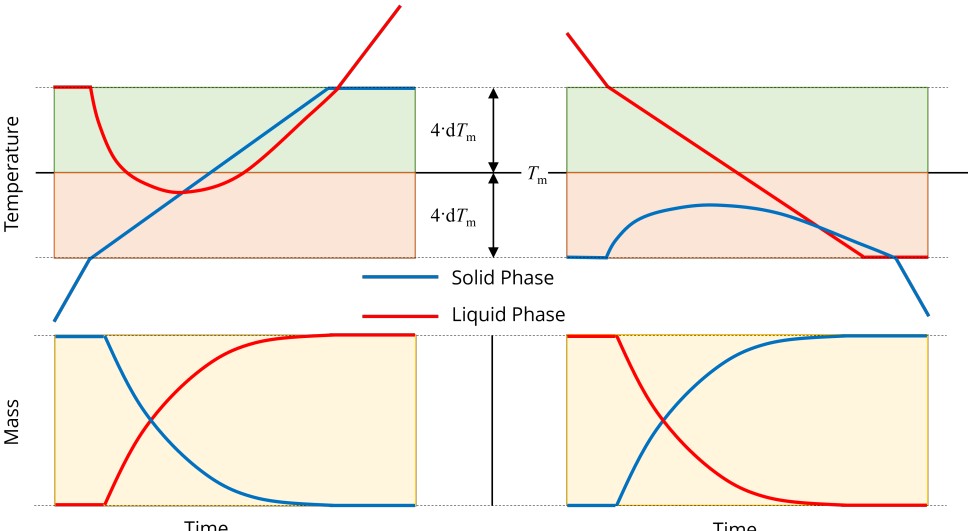

**Figure 2.** Illustration of (**left**) melting and (**right**) freezing. $T_{\mathrm{m}}$ indicates the steel melting temperature and $dT_{\mathrm{m}}$ is a constant used for calculating the rates of melting and freezing.

### 2.5. Arc Efficiency

A novel model for electrical energy efficiency based on arc visibility has been implemented. Observations from process data and literature indicate that arc efficiency, defined as the percentage of supplied electrical energy that heats steel, changes during the course of a heat [21]. While visibility of the arc is just one of many factors that affects arc efficiency [21], a description of arc coverage is a natural extension of the dynamic model states and can provide valuable insight into the current and future states of the process.

Figure 3 denotes the dimensions used by the arc efficiency model. Depending on the electrode height and the heights of the scrap (solid), liquid and slag phases, the electric arc length $l_{\mathrm{arc}}$ can be completely covered or partially exposed. The length of the electric arc $l_{\mathrm{arc}}$ is a constant while the position of the arc in the furnace is determined by the height of the electrode $h_{\mathrm{electrode}}$, which changes during the boredown and melting stages of the process. The heights of the phases and the total bath are given by:

$$h_{(\mathrm{solid/liquid/slag})} = \frac{m_{(\mathrm{solid/liquid/slag})}}{A_{\mathrm{furnace}}\rho_{(\mathrm{solid/liquid/slag})}}$$

$$h_{\mathrm{bath}} = h_{\mathrm{scrap}} + h_{\mathrm{liquid}} + h_{\mathrm{slag}}. \tag{14}$$

The visibility of the arc can then be written:

$$vis_{\mathrm{arc}} = \max\left(\min\left(\frac{h_{\mathrm{electrode}} - h_{\mathrm{bath}}}{l_{\mathrm{arc}}}, 1\right), 0\right). \tag{15}$$

The arc visibility is zero when the full length of the arc is below the cumulative height of the bath components (scrap $h_{\mathrm{scrap}}$, liquid $h_{\mathrm{liquid}}$ and slag $h_{\mathrm{slag}}$). The visibility is combined with additional model parameters in order to write the model for arc energy losses that includes estimation parameter $k_{\mathrm{loss}}$, which can be used to tune the model to better follow individual heats:

$$x_{\mathrm{arc}}^{\mathrm{loss}} = k_{\mathrm{loss}}\left(vis_{\mathrm{arc}} + k_{\mathrm{basket}}^{\mathrm{loss}}\right). \tag{16}$$

Using process data to calculate the electrode height $h_{\mathrm{electrode}}$ is challenging because the electrodes are consumed during the process, causing the length of the electrodes to vary from heat to heat. Instead of using process data to determine to electrode position, the

electrode height is modeled with an equation that captures typical boredown behavior as a function of the total instantaneous solid and liquid mass and the total electrical energy supplied to the furnace kWh:

$$h_{\text{electrode}} = \exp(-\text{kWh}/\text{kWh}_{\text{basket}}) + \frac{m_{\text{solid}} + m_{\text{liquid}}}{A_{\text{furnace}}\rho_{\text{liquid}}}. \tag{17}$$

The density of slag $\rho_{\text{slag}}$ is modeled as a function of the overall liquid fraction, consistent with the observation that foaminess increases as meltdown progresses [22]:

$$\rho_{\text{slag}} = 120 + 1380 \exp(-x_1/x_{\text{basket}}). \tag{18}$$

Equations (16)–(18) each include a term with the subscript *basket*. These terms are model-fitting constants that are fit for the cases of one-, two- and three-basket heats.

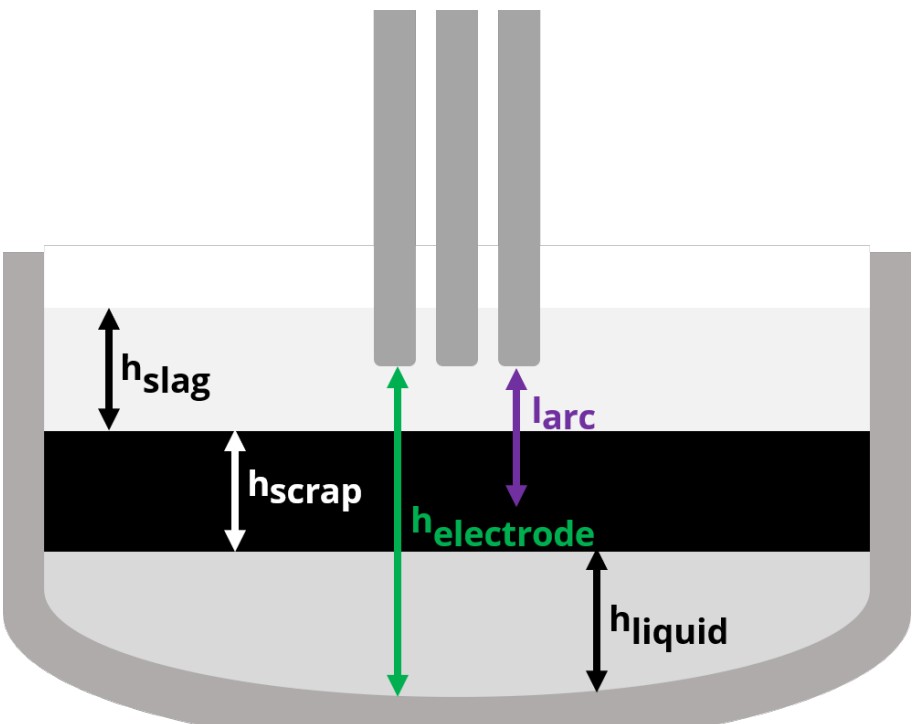

**Figure 3.** Variable and static dimensions used to calculate arc visibility for the arc efficiency model.

## 3. Results and Discussion

### 3.1. Model Behavior

Figure 4 shows typical model meltdown behavior. Heats typically result in the production of approximately 140 tonnes of liquid steel. The inner solid mass is heated directly by the electric arc and always begins to melt first, while the outer solid mass begins to melt before the inner solid is completely liquefied. The melting process is interrupted by pauses in process operation and the addition of the second basket, at which point all mass solidifies before continuing to be heated and re-melted. Each time a new heat begins, the furnace is emptied (solid steel, liquid steel, slag and component masses are reset to initial states, along with all temperatures except for the furnace roof and vessel).

The process begins when the power and gas burners are turned on. The electrodes are located in the center of the furnace and supply power to the most closely situated charged material. The arc power is therefore used to heat the inner steel control volume as described in Appendix A.1. As a result, the inner solid temperature rises much faster than the outer solid temperature. Heat transfer limitations described in Section 2.3 govern the rate at which arc energy is dissipated from the inner to the outer control volume. The furnace

surroundings are heated by energy losses from the burners and arc as well as by heat transfer from the steel. Eventually, the inner solid becomes hot enough to melt and an inner liquid mass begins to appear. At this point in the process, additional planned baskets will usually be added, cooling the furnace contents. Time delays in adding baskets also cause the furnace contents to cool undesirably. Power continues to be supplied to the furnace, and the outer solid finally begins to melt before the inner solid has fully disappeared.

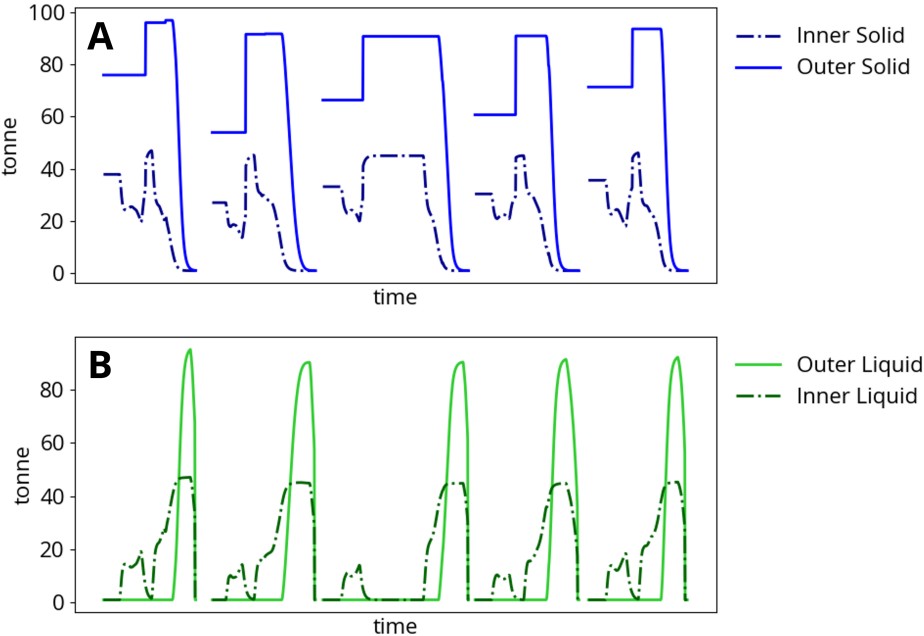

**Figure 4.** Steel meltdown dynamics and change in (**A**) solid mass and (**B**) liquid mass during five consecutive two-basket heats.

Figure 5 shows model agreement with process data from 250 heats for liquid steel temperature and weight after tapping. The model does not account for a hot heel, which can vary from heat to heat and may explain some of the observed scatter in weight agreement. The hot heel discrepancy may also affect the scatter in temperature agreement. Both the weight and temperature agreement are within reasonable expectations for model behavior and measurement accuracy.

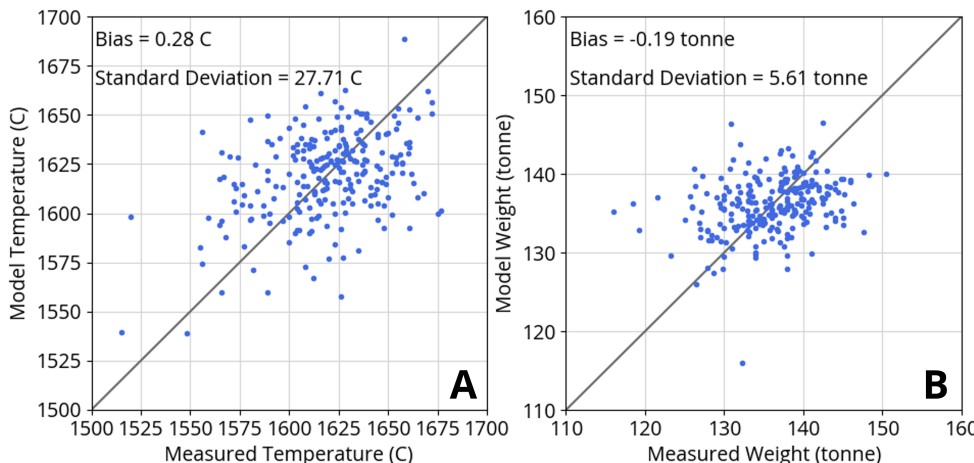

**Figure 5.** Agreement of model predictions with process measurements from 250 heats for (**A**) liquid steel temperature and (**B**) liquid steel weight after tapping. Model biases for temperature and weight are negligible while residual standard deviations reflect scatter.

### 3.2. Recursive Estimation of Arc Efficiency

In order for MPC to be effective, the underlying model describing the process needs to be able to accurately predict the behavior of the optimization targets as the heat progresses. For the purpose of optimization for energy savings, the model not only has to follow the present efficiency, but must also be able to predict the future efficiency with sufficient accuracy. The goal of developing the arc energy loss model described in Section 2.5 is therefore to accurately predict efficiency at upcoming stages in the meltdown process such that the power profile can be optimized accordingly. Arc power can be adjusted down and up during low and high efficiency periods, respectively, in order to minimize energy losses to the environment.

The arc efficiency model includes basket-dependent constants as described in Section 2.5. In order to fit the constants $k_{\mathrm{basket}}^{\mathrm{loss}}$, kWh$_{\mathrm{basket}}$ and $x_{\mathrm{basket}}$, a precursor study using process data was performed in which the percentage of arc power loss to the furnace roof and vessel ($x_{\mathrm{arc}}^{\mathrm{loss}}$ in Equation (16)) was recursively estimated with a KF [13]. The average result of this study for one-, two- and three- basket heats is shown in Figure 6, where a higher arc energy loss proportion corresponds to a lower arc efficiency. The discernible features denoted in Figure 6 allow us to propose a physical explanation that we can later use to model arc efficiency. These features are most apparent for one-basket heats, as multiple-basket heat results are impacted by the variable proportions of electric energy added per basket. High arc losses at the beginning of the heats are explained by arc exposure during boredown. As boredown continues, the arc is covered by scrap and the loss proportion decreases until the scrap begins to melt and re-exposes the arc. The arc is, however, only temporarily exposed, as the appearance of a liquid steel phase is followed quickly by a liquid slag phase, and the reactions between liquid steel and slag components lead to a larger, foamier slag phase. This foamy slag phase provides the arc with coverage as the heat nears completion.

Efficiency model constants $k_{\mathrm{basket}}^{\mathrm{loss}}$, kWh$_{\mathrm{basket}}$ and $x_{\mathrm{basket}}$ are fit to best reproduce the average data for one-, two- and three-basket heats presented in Figure 6. The arc efficiency model effectively reproduces the one- and two-basket heat data, but performs less well when compared to the three-basket heat data. The discrepancy between the model and the three-basket heat data could be due to poor statistics, as far fewer three-basket heats were recorded in the data series.

While the basket-dependent parameters do enable the model to capture average process behavior, there still exists a variable degree of energy losses between heats with the same number of baskets. These differences between heats could be due to many factors not currently accounted for by the efficiency model, including variable density of baskets, electrode tip wear and hot heel. The strategy employed for addressing this variation is to include scaling parameter $k_{\mathrm{loss}}$ in the model for arc efficiency given by Equation (16). Unlike the efficiency model constants, which are tuned for all heats with the same number of planned charged baskets, the $k_{\mathrm{loss}}$ term in Equation (16) is a designated parameter for recursive estimation. Variations in efficiency from heat to heat are to be expected and can be followed using a KF [13]. Figure 7 shows an example of the impact of recursive estimation on the same five consecutive two-basket heats plotted in Figure 4. The ballistic simulation uses a $k_{\mathrm{loss}}$ value that best matches the average for all heats with the same number of baskets. The ballistic simulation follows the trend in cooling water process data well, but is prone to sometimes overpredicting heat losses and the resulting cooling water outlet temperature. Recursive estimation of $k_{\mathrm{loss}}$ fares better: after overcoming initial errors that come from reinitializing the heat, the estimation results match both the trend and level of cooling water temperature data.

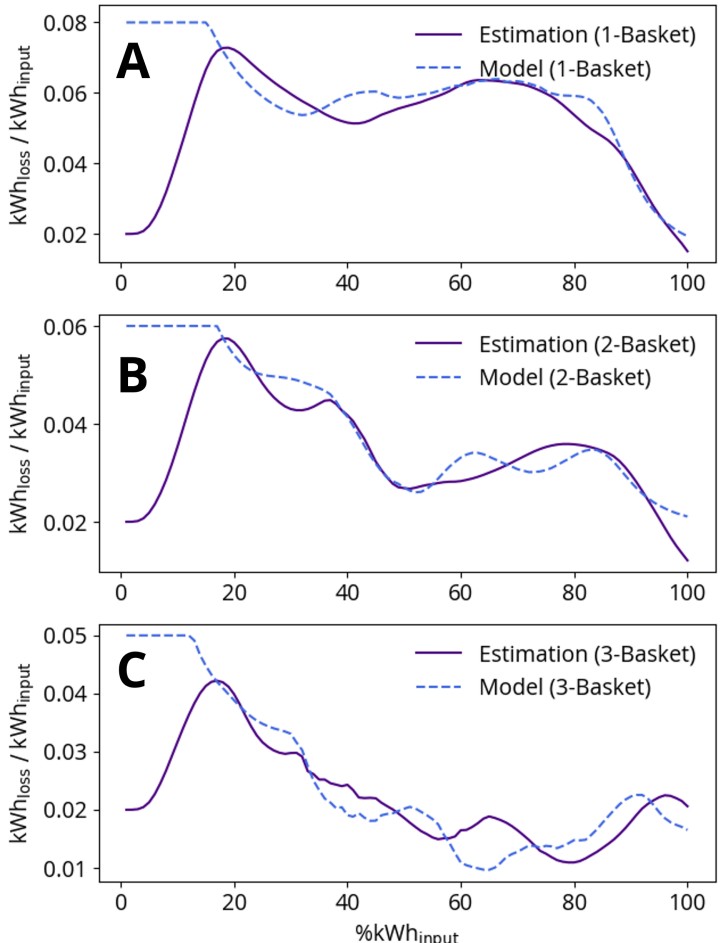

**Figure 6.** Comparison of arc efficiency estimation and model for (**A**) one-basket heats, (**B**) two-basket heats and (**C**) three-basket heats. The *x*-axis is the normalized progress of each heat as measured by the percentage of total electric energy added to the furnace. Low %kWh$_{\text{input}}$-results are not meaningful as the efficiency estimation requires several samples to change from the initial guess factor of 0.02.

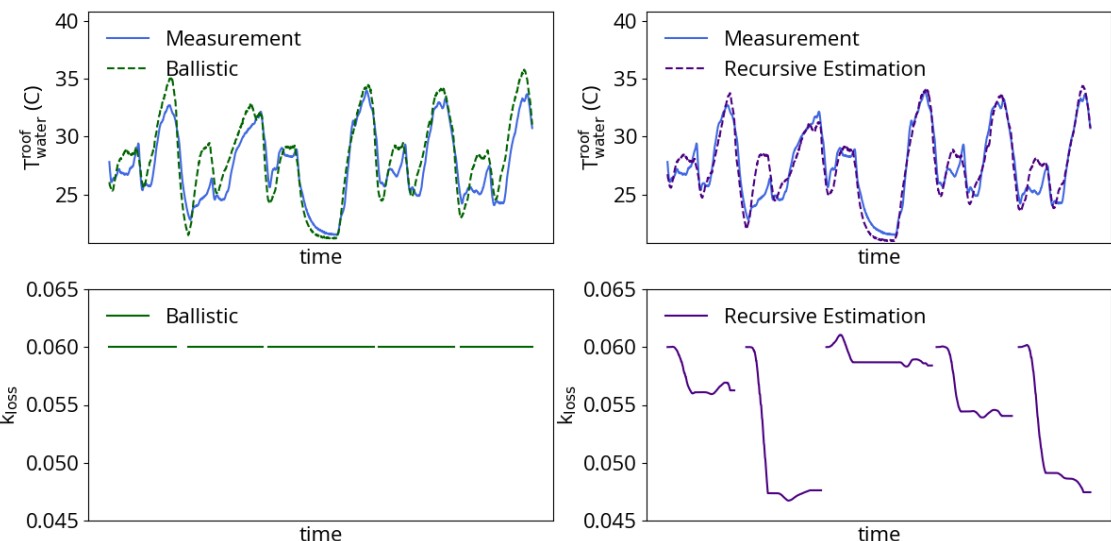

**Figure 7.** Example of arc loss coefficient $k_{\text{loss}}$ estimation during five consecutive two-basket heats. (**left**) A constant $k_{\text{loss}}$ produces model results that follow the cooling water dynamics but sometimes overpredict the outlet temperature. (**right**) Estimation of $k_{\text{loss}}$ produces more accurate model results.

### 3.3. Industrial Use and Application

Model predictive control (MPC) is an advanced method of process control that determines a sequence of process inputs that optimizes a predicted process output at specified time points in the future [23]. Online MPC routines re-evaluate the current process state at each successive sampling time, allowing the optimization to adapt to process disturbances. The process control scheme referred to in this work can be more specifically described as Non-linear model predictive control (NMPC) because the predicted response to proposed inputs are calculated based on a non-linear process model.

MPC simulations were performed based on logged plant data from 250 heats with the Cybernetica$^{\text{TM}}$ Cenit software. The goal of these MPC simulations is to optimize the electrical power input in order to increase the overall efficiency of the arc power. In order to adapt the process data for the MPC study, the basket contents and schedule of charges are preserved according to logged data without the exact schedule being preemptively revealed to the MPC. Logged power input is overwritten by the closed-loop simulation. Because the operation of the gas burners should be in sync with the accumulated electrical energy added to the furnace, logged LNG and oxygen flows are replaced with the gas burner recipe used in plant operation. Logged time delays and pauses in electrical power supply are preserved in the MPC simulations.

In the Cybernetica$^{\text{TM}}$ Cenit implementation of MPC, the optimization takes the form of minimizing an objective function. Because the EAF is operated as a batch process, the process outputs that contribute to the objective function are evaluated at the end of the batch (the time at which the model predicts the furnace contents are fully melted). The MPC algorithm seeks to simultaneously minimize the total batch time and maximize the efficiency of the electric arc based on the objective function $J$:

$$J = \frac{1}{2}\Delta U^{\text{T}} \mathbf{S} \, \Delta U + R^{\text{T}}(Z - Z_{\text{max}}). \tag{19}$$

The optimization criteria $J$ is a scalar calculated from the sum of the right-hand side terms in Equation (19). $U$ represents manipulated variable (MV) process inputs and $\Delta U$ is the vector of changes to manipulated inputs proposed by the optimization. Changes to MVs are weighted by the penalties contained by the diagonal of matrix $\mathbf{S}$. $Z$ is vector of control variables (CV) calculated by the model, and $Z_{\text{max}}$ is a vector of soft maximum constraints for each CV. The violation of each constraint is weighted linearly by vector $R$, and the term is only evaluated for the largest constraint violation in the prediction horizon.

For the EAF optimization simulations, the elements of the $U$-vector are:

- $U_{1-15}$ or MV$_{1-15}$: Shift from the nominal power profile during optimization interval ($i = 1$ to 15) (MW).

The MPC optimizes deviation $U$ from a nominal power profile in order to propose a more efficient power profile solution. The MV in this optimization problem is therefore change to power input rather than the power input itself. The prediction horizon is divided into 15 intervals, each of which is shifted by a separate $U_i$. The nominal power profile is given by plant recipes for one-, two- and three-basket heats. The elements of the $Z$-vector are:

1. $Z_1$ or CV$_1$: Batch time (seconds)
2. $Z_2$ or CV$_2$: Energy losses from the electric arc (kWh).

The elements of $Z_{\text{max}}$ are in this case set to zero in order to direct the optimization to minimize both of the CVs.

Because the arc efficiency changes as the solid scrap melts down and a slag phase forms, as discussed in Section 3.2, adjusting the power levels over the course of the heat can lead to a more optimal power profile. Figure 8 shows an illustration of power profile optimization. The MPC scheme uses a finite receding horizon, meaning that the as the heat proceeds the output power profile will have shorter remaining duration. The criteria for ending the output power is the model prediction of full meltdown of the solid mass

in the furnace. The example batches following the nominal and optimized input power trajectories are predicted to end at different times, and the optimized trajectory is projected to incur fewer energy losses directly from the arc to the environment.

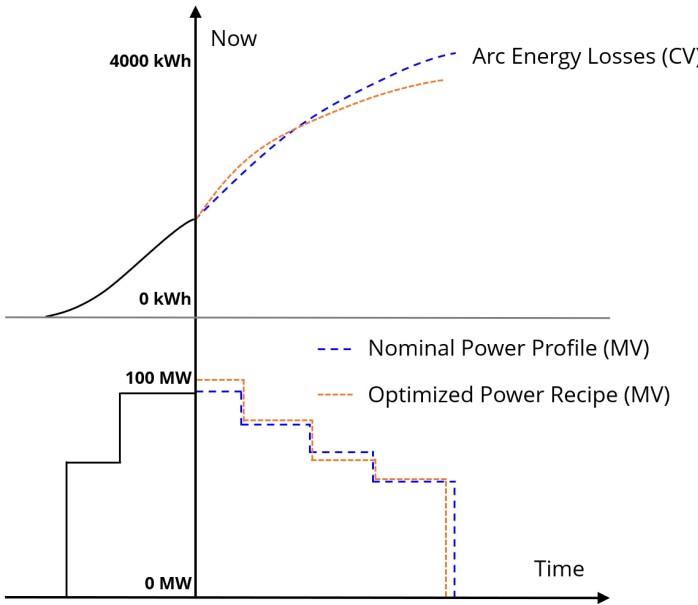

**Figure 8.** An illustration of MPC optimization for batch time and electric arc efficiency. Historical MV and model CV predictions are shown as solid black lines. Within the prediction horizon, optimized MV (MPC output) and CV are shown in orange, while nominal MV and the resulting CV are shown in blue.

Figure 9 shows the energy savings predicted by the MPC simulations for one-, two- and three-basket heats. The average predicted savings are:

- One-basket heats: 15.75 kWh/tonne per heat (2315 kWh per heat, based on an average charged weight of 147 tonne);
- Two-basket heats: 13.32 kWh/tonne per heat (1945 kWh per heat, based on an average charged weight of 146 tonne);
- Three-basket heats: 6.78 kWh/tonne per heat (983 kWh per heat, based on an average charged weight of 145 tonne).

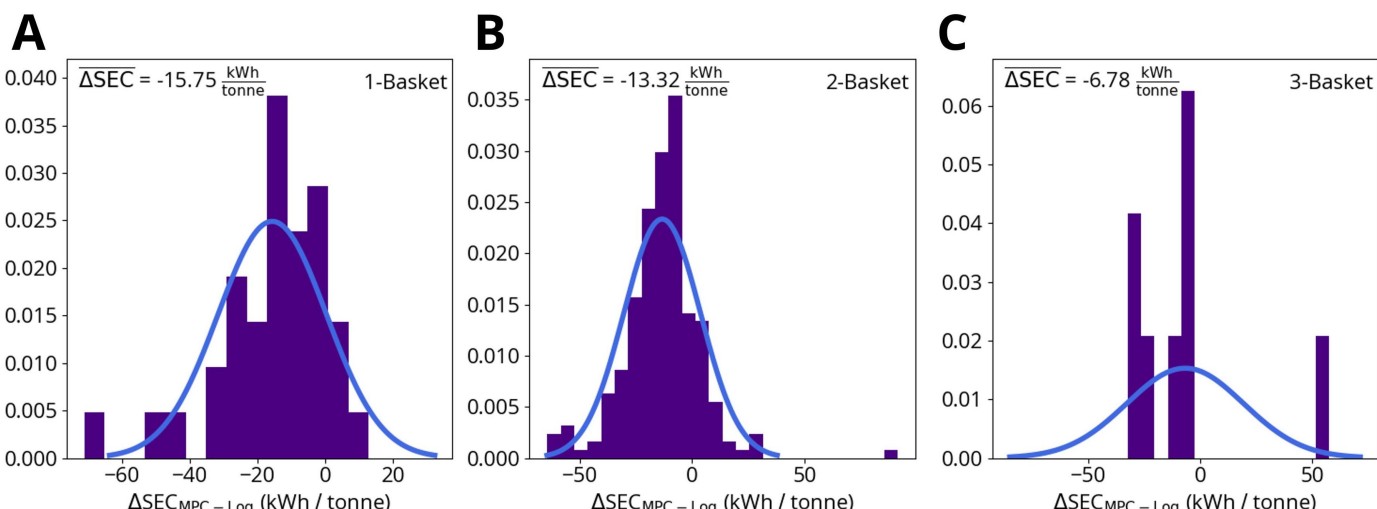

**Figure 9.** Change in specific energy consumption (SEC) due to MPC for (**A**) one-basket heats, (**B**) two-basket heats and (**C**) three-basket heats. ΔSEC is calculated by subtracting the logged SEC up to the point where the model predicts steel meltdown from the MPC simulation result.

These results only account for the process energy savings up to the point when the model predicts that all the scrap metal has melted. This means that if the model predicts that the logged power profiles continue to heat the furnace after all the scrap metal is liquefied, which happens frequently, the additional energy savings beyond the point of full meltdown are neglected. While the model predicts that accounting for turning off the furnace earlier can cut down energy usage per heat by up to 15 kWh/tonne, these additional savings are not included in the average predicted savings listed above or in the data shown in Figures 9 and 10. The purpose of this savings criteria is to evaluate the MPC scheme's potential for efficiency improvements independently from the model's accuracy of end-point prediction.

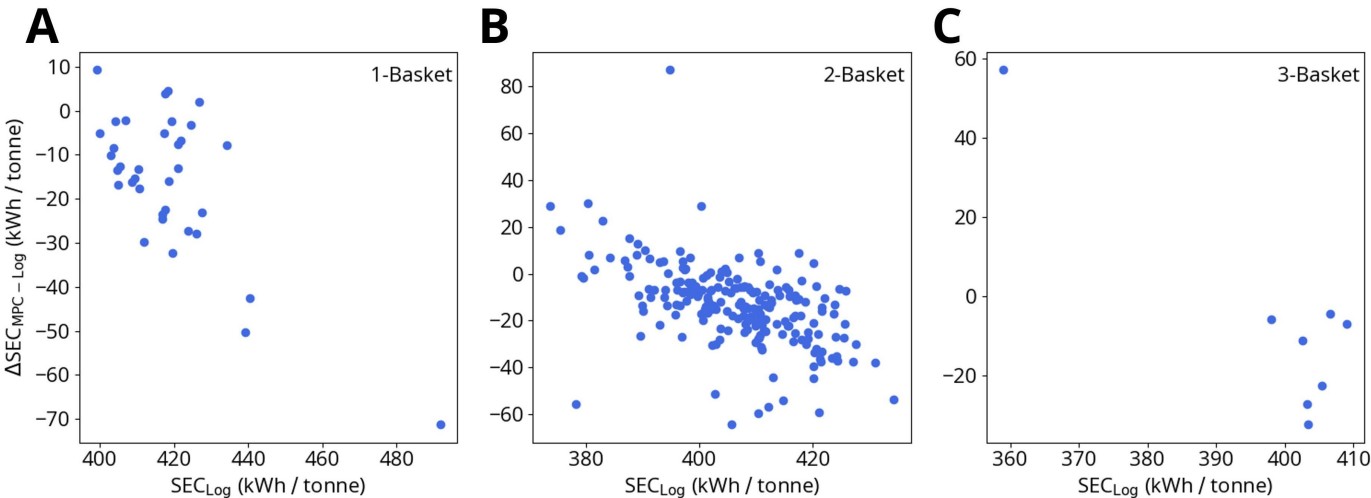

**Figure 10.** Logged specific energy consumption (SEC) up to the point of melting vs. change in SEC due to MPC for (**A**) one-basket heats, (**B**) two-basket heats and (**C**) three-basket heats. ΔSEC is calculated as described in Figure 9.

While the MPC simulations predict the largest energy savings for one-basket heats, the overall findings for one-, two- and three-basket heats are similar. Within these subgroups, the electrical energy savings are not uniform for all heats: Figure 10 shows that the MPC savings (ΔSEC) are correlated with the magnitude of the logged specific electrical energy consumption (SEC). As SEC approaches approximate threshold values of 400 kWh/tonne (one-basket heats) and 390 kWh/tonne (two- and three-basket heats) ΔSEC steadily decreases, potentially indicating that the MPC is not able to save significant amounts of energy beyond a given lower bound.

The decreased energy demand for melting the furnace contents while following MPC recommendations raises the question of whether the the optimized power profile significantly affects the endpoint state of the liquid steel. The process and model dynamics are such that fully melting the scrap metal within the time frame of a standard batch requires heating the liquid steel well above the melting temperature. In practice, temperature measurements are made only after tapping, making a temperature target difficult to verify, and the assumption that typical meltdown temperatures are high enough for downstream processing has been shown to be sufficient. Because the optimization scheme uses a meltdown criteria rather than an explicit temperature target, the energy-saving recommendations could conceivably result in lower temperatures that are not optimal for further processing and refining of the steel. To investigate this question, model predictions of liquid steel temperatures from optimized and logged power profile simulations are examined and compiled in Table 4. Two process data-derived temperatures are presented: $T_{Log}^{Melt}$ is the liquid steel temperature at the moment the model predicts the steel is melted, while $T_{Log}^{Full}$ is the liquid steel temperature when the logged power is shut off. Because the MPC simulations shut off power once all the steel has melted, $T_{Log}^{Melt}$ is the most appropriate quantity for comparison to $T_{MPC}$.

**Table 4.** Temperature of fully-liquefied steel prior to tapping as predicted by the model when following logged process data and MPC simulations. Both means and standard deviations are presented.

| | $T_{Log}^{Full}$ | $T_{Log}^{Melt}$ | $T_{MPC}$ |
|---|---|---|---|
| **1-Basket** | 1688.5 °C $\pm$ 27.8 °C | 1684.5 °C $\pm$ 27.9 °C | 1693.0 °C $\pm$ 13.0 °C |
| **2-Basket** | 1693.3 °C $\pm$ 21.6 °C | 1687.6 °C $\pm$ 23.4 °C | 1703.7 °C $\pm$ 17.1 °C |
| **3-Basket** | 1717.9 °C $\pm$ 2.1 °C | 1717.2 °C $\pm$ 2.2 °C | 1702.5 °C $\pm$ 0.9 °C |

The results presented in Table 4 show that, at the point of meltdown, the liquid steel actually reaches higher temperatures in the MPC simulations than in the simulations with logged data. These results point to an interesting finding: according to the model, the optimized power profile leads not only to lower process energy demand due to reduced heating of the environment directly by the electric arc, but also to more efficient heating of the steel by the energy that is able to be absorbed. MPC meltdown temperatures are also higher than predicted temperatures at the point of logged power shut off, indicating that following the optimized profile should not cause the liquid steel to be too cold at tapping such as to cause problems for downstream plant processes.

## 4. Conclusions

The EAF is a challenging process to model and optimize from a precision point of view: there are significant uncertainties associated with materials and electrodes that can be difficult to resolve using an automated approach. This work aimed at formulating a first-principles mathematical model in terms of ordinary differential equations for the state variables of the EAF that can be adapted to process data using recursive parameter estimation. The resulting model described in this article is of manageable size yet sufficiently detailed and adaptable to be useful for process optimization.

An MPC-based optimization application based on this model has been running online using data for a 140 tonne EAF furnace since August 2020, demonstrating that the model is fast enough for industrial deployment. The predicted metal temperatures and weights were found to be in reasonably good agreement with the measured values. Results indicate that MPC-based process operation leads to both a reduction in total energy usage as well as more efficient dissipation and heating of the steel by the consumed energy.

While several MPC studies for EAF processes have been reported in the literature [3,6], this study is, to the best of the authors' knowledge, the first where the efficiency of the power input and electric arc has been the target for MPC optimization. The framework for optimizing power input with MPC opens up the possibility of incorporating a more complex model of the electrical power in order to achieve more effective and sensitive optimization of arc efficiency [21]. Additional optimization scenarios can be considered in further work: an economic optimization of burner vs. electrical power can be implemented using the same framework, and new data and dynamic states can be added to the model in order to optimize for operational costs such as equipment life cycle and electrode consumption [24].

**Author Contributions:** Conceptualization, S.O.W., A.H., N.H., V.-V.V. and M.S.; methodology, S.J., S.O.W., V.-V.V. and M.S.; software, S.J., S.O.W. and A.H.; validation, S.J., V.-V.V. and N.H.; formal analysis, S.J.; investigation, S.J. and N.H.; resources, S.O.W. and N.H.; data curation, S.J. and S.O.W.; writing—original draft preparation, S.J.; writing—review and editing, S.O.W., V.-V.V. and M.S.; visualization, S.J. and A.H.; supervision, S.O.W. and V.-V.V.; project administration, S.O.W. and V.-V.V.; funding acquisition, S.O.W. and M.S. All authors have read and agreed to the published version of the manuscript.

**Funding:** This research was funded by the European Commission, Directorate-General for Research and Innovation, through the project "Model-based optimisation for efficient use of resources and energy" (MORSE), grant number 768652. https://www.spire2030.eu/morse (accessed on 26 September 2021).

**Institutional Review Board Statement:** Not applicable.

**Informed Consent Statement:** Not applicable.

**Data Availability Statement:** Restrictions apply to the availability of these data. The data presented in this study are available on request from the corresponding author with the permission of authors/partners at Outokumpu Stainless Oy.

**Acknowledgments:** The authors acknowledge helpful discussions with Eero Auer and Esa Puukko from Outokumpu regarding the collection, processing and results of EAF data. The tragic death of Ralf Pierre, a much appreciated expert in EAF modeling and very helpful partner in many fruitful discussions, has left the authors deeply saddened.

**Conflicts of Interest:** The authors declare no conflict of interest. The funders had no role in the design of the study; in the collection, analyses, or interpretation of data; in the writing of the manuscript, or in the decision to publish the results.

## Abbreviations

The following abbreviations are used in this manuscript:

| | |
|---|---|
| EAF | Electric Arc Furnace |
| MPC | Model Predictive Control |
| NMPC | Non-linear Model Predictive Control |
| SEC | Specific Energy Consumption |
| KF | Kalman Filter |
| LNG | Liquefied Natural Gas |
| VF | View Factor |
| CV | Controlled Variable |
| MV | Manipulated Variable |
| DRI | Direct Reduced Iron |

## Nomenclature

The nomenclature for constants and variables used in the main text and appendices of this manuscript are listed:

| | | |
|---|---|---|
| $A_{ij}$ | Area for heat transfer between mass $i$ and mass $j$ | $\mathrm{m}^2$ |
| $d_i$ | Diameter of $i$ | m |
| $\frac{\mathrm{d}}{\mathrm{d}t}$ | Derivative operator | |
| $\epsilon_j$ | Radiation emissivity of surface $j$ | |
| $F_i$ | Molar rate of change of component $i$ | $\frac{\mathrm{mol}}{\mathrm{s}}$ |
| $C_{\mathrm{p},\,i}$ | Heat capacity of $i$ | $\frac{\mathrm{J}}{\mathrm{kg\,K}}$ |
| $h_k$ | Height of mass $k$ | m |
| $H_i$ | Enthalpy of component $i$ | $\frac{\mathrm{J}}{\mathrm{kg}}$ |
| $k_{ij}$ | Heat transfer coefficient between type $i$ and type $j$ | $\frac{\mathrm{W}}{\mathrm{m}^2\mathrm{K}}$ |
| $m_i^j$ | Mass of component $i$ in phase j | kg |
| $M_i$ | Molar mass of component $i$ | $\frac{\mathrm{g}}{\mathrm{mol}}$ |
| $Q_{ij}$ | Heat flowing from mass $i$ to mass $j$ | W |
| $P_k$ | Power from source $k$ | MW |
| $p_i$ | Pressure of phase $i$ | Pa |
| $r_i$ | Reaction rate of reaction $i$ | $\frac{\mathrm{kg}}{\mathrm{s}}$ |
| $r_{\mathrm{particle}}$ | Radius of particle | m |
| $\rho_i$ | Density of $i$ | $\frac{\mathrm{kg}}{\mathrm{m}^3}$ |
| $\sigma_{SB}$ | Stefan-Boltzmann constant for radiation | $5.670374 \times 10^{-8}\,\frac{\mathrm{W}}{\mathrm{m}^2\mathrm{K}^4}$ |
| $t$ | Time | s |
| $T_j$ | Temperature of phase $j$ | K |
| $v_i$ | Stoichiometric coefficient of component $i$ in a chemical reaction | |
| $V_k$ | Volume of mass $k$ | $\mathrm{m}^3$ |

| $W_i$ | Mass rate of change of component $i$ | $\frac{\text{kg}}{\text{s}}$ |
| $x_i^j$ | Mass fraction of component $i$ in phase $j$ | |
| $x_j$ | Area fraction of control volume $j$ | |
| subscript: $b$ | Liquid slag | |
| subscript: $c$ | Solid slag | |
| subscript: $g$ | Gas | |
| subscript: $l$ | Liquid steel | |
| subscript: $r$ | Roof | |
| subscript: $s$ | Solid steel | |
| subscript: $v$ | Vessel | |

## Appendix A. Extended Model Details

*Appendix A.1. Electrical and Chemical Power*

The power input supplied by the electric arc $P_{\text{arc}}$ is a logged process input. The arc power is partitioned between several model masses:

$$P_{\text{arc}}^{\text{gas}} = x_{\text{arc}}^{\text{gas}} P_{\text{arc}}$$
$$P_{\text{arc}}^{\text{loss}} = x_{\text{arc}}^{\text{loss}} P_{\text{arc}} \tag{A1}$$
$$P_{\text{arc}}^{\text{steel}} = \left(1 - x_{\text{arc}}^{\text{gas}} - x_{\text{arc}}^{\text{loss}}\right) P_{\text{arc}}$$

The calculation and estimation of $x_{\text{arc}}^{\text{loss}}$ is described in Section 2.5, while $x_{\text{arc}}^{\text{gas}}$ is a model constant for the fraction of arc power used to heat the gas phase. $P_{\text{arc}}^{\text{loss}}$ is used to heat two pieces of furnace equipment: the furnace vessel and roof. The further partitioning of $P_{\text{arc}}^{\text{loss}}$ is determined by model constant $x_{\text{v}}^{\text{loss}}$:

$$P_{\text{arc}}^{\text{vessel}} = x_{\text{vessel}}^{\text{loss}} P_{\text{arc}}^{\text{loss}}$$
$$P_{\text{arc}}^{\text{roof}} = \left(1 - x_{\text{vessel}}^{\text{loss}}\right) P_{\text{arc}}^{\text{loss}} \tag{A2}$$

The molar LNG flow rate into the furnace $F_{LNG}$ is given by conversion from logged gas flow rates according to standard temperature and pressure (STP; $T = 0\,^{\circ}$C, $p = 100$ kPa). The molar flow rate of oxygen $F_{O_2}$ is similarly calculated by conversion from logged standard flow rates. Given that enough oxygen is present, the LNG is assumed to combust completely, allowing us to write an expression for the LNG power $P_{\text{LNG}}$:

$$P_{\text{LNG}} = \left(1.0 \times 10^{-3}\right) \min\left(F_{LNG}, \frac{F_{O_2}}{2}\right) M_{\text{LNG}} H_{\text{LNG}} \tag{A3}$$

$M_{\text{LNG}}$ and $H_{\text{LNG}}$ refer to the the molar mass and enthalpy of combustion for the specified composition of LNG. $P_{\text{LNG}}$ heats both the steel and gas phases, with the gas phase being heated to a greater extend as meltdown progresses:

$$x_{\text{LNG}}^{\text{gas}} = 0.25\left(\tanh\left(5x_{\text{liquid}}^{\text{overall}} - 2.5\right) + 2\right)$$
$$P_{\text{LNG}}^{\text{gas}} = x_{\text{LNG}}^{\text{gas}} P_{\text{LNG}} \tag{A4}$$
$$P_{\text{LNG}}^{\text{steel}} = \left(1 - x_{\text{LNG}}^{\text{gas}}\right) P_{\text{LNG}}$$

Oxygen consumption by the combustion of LNG can be calculated:

$$F_{O_2}^{\text{LNG}} = 2 \times \min\left(F_{LNG}, \frac{F_{O_2}}{2}\right) \tag{A5}$$

Any oxygen not consumed by LNG combustion remains available for further chemical reactions. Oxygen not consumed by LNG can be used for combustion with CO:

$$F_{O_2}^{CO} = \min\left(F_{O_2} - F_{O_2}^{LNG}, \frac{k_{CO}n_{CO}}{2}\right) \tag{A6}$$

$n_{CO}$ denotes the accumulated moles of CO from refining reactions present in the furnace and $k_{CO}$ refers to a model limiting rate constant for the CO combustion reaction. The power released by CO combustion is given:

$$p_{CO} = F_{O_2}^{CO} H_{CO} \tag{A7}$$

$H_{CO}$ refers to the enthalpy of combustion for CO. The amount of oxygen available for refining reactions can be calculated based on the mass of liquid steel species:

$$m_{refine}^{inner} = \sum_{i = Fe, C, Si, Cr}^{inner} m_{i, liquid}^{inner}$$

$$m_{refine}^{outer} = \sum_{i = Fe, C, Si, Cr}^{outer} m_{i, liquid}^{outer} \tag{A8}$$

$$F_{O_2}^{refine} = \max\left(F_{O_2} - F_{O_2}^{LNG} - F_{O_2}^{CO}, 0\right)$$

$m_{i, liquid}$ denotes the mass of species $i$ within a given control volume and $k_{refine}$ refers to a model limiting rate constant of refining for the liquid steel components. $F_{O_2}^{refine}$ is partitioned to the inner and outer control volumes and used for the reactions described in Appendix A.2:

$$F_{O_2}^{refine, inner} = \frac{m_{refine}^{inner}}{m_{refine}^{inner} + m_{refine}^{outer}} F_{O_2}^{refine}$$

$$F_{O_2}^{refine, outer} = \frac{m_{refine}^{outer}}{m_{refine}^{inner} + m_{refine}^{outer}} F_{O_2}^{refine} \tag{A9}$$

The power from all refining reactions is calculated from the sum of reactions described in Appendix A.2:

$$P_{refine} = \sum_{r_{oxygen}} \left(F_{O_2, inner}^{r_{oxygen}} + F_{O_2, outer}^{r_{oxygen}}\right) H_{O_2}^{r_{oxygen}} + \sum_{r_{equilibrium}} \left(F_{XO, inner}^{r_{equilibrium}} + F_{XO, outer}^{r_{equilibrium}}\right) H_{XO}^{r_{equilibrium}} \tag{A10}$$

$H_{O_2}$ refers to the enthalpy of each refining reaction involving oxygen and $H_{XO}$ refers to the enthalpy of each steel-slag equilibrium reaction per *mol* of **XO**, where **XO** is the oxide species listed for each reaction in Tables A3 and A4.

The total chemical power to the steel is given:

$$P_{chemical} = P_{LNG}^{steel} + P_{refine} + P_{CO} \tag{A11}$$

The total steel-heating power is partitioned between the inner and outer control volumes, with all of $P_{arc}^{steel}$ being used to heat the inner steel:

$$P_{inner} = x_{inner}\left(P_{LNG}^{steel} + P_{refine} + P_{CO}\right) + P_{arc}^{steel}$$

$$P_{outer} = x_{outer}\left(P_{LNG}^{steel} + P_{refine} + P_{CO}\right) \tag{A12}$$

$x_{inner}$ and $x_{outer}$ are the area fractions of the inner and outer control volumes, respectively, as calculated from the model dimensions given in Appendix A.4. The total power to the gas phase is given:

$$P_{gas} = P_{LNG}^{gas} + P_{arc}^{gas} \tag{A13}$$

*Appendix A.2. Reaction Kinetics*

The rates of oxygen consumption by the different reactions listed in Equation (2) depend on the species mass fractions. The rates of consumption and production of the other species involved can be calculated from the oxygen consumption rate. The consumption and production rates of different species are given in Table A1 and hold for both the inner and outer control volumes, where the steel mass fractions and oxygen available for refining can vary.

**Table A1.** Oxygen consumption and species rate of change for each steel reaction with oxygen. $F$ denotes molar rates of change, $v$ denotes stoichiometric coefficients and $x$ denotes mass fractions.

| Reaction | O$_2$ Consumption Rate | Species $i$ Rate of Change |
|---|---|---|
| $Fe + \frac{1}{2}\{O_2\} \rightarrow (FeO)$ | $F_{O_2}^{Fe \rightarrow FeO} = x_{Fe}^{steel} F_{O_2}^{refine}$ | $F_i^{Fe \rightarrow FeO} = \frac{v_i^{Fe \rightarrow FeO}}{v_{O_2}^{Fe \rightarrow FeO}} F_{O_2}^{Fe \rightarrow FeO}$ |
| $[C] + \frac{1}{2}\{O_2\} \rightarrow \{CO\}$ | $F_{O_2}^{C \rightarrow CO} = \frac{1}{2} x_C^{steel} F_{O_2}^{refine}$ | $F_i^{C \rightarrow CO} = \frac{v_i^{C \rightarrow CO}}{v_{O_2}^{C \rightarrow CO}} F_{O_2}^{C \rightarrow CO}$ |
| $[C] + \{O_2\} \rightarrow \{CO_2\}$ | $F_{O_2}^{C \rightarrow CO_2} = \frac{1}{2} x_C^{steel} F_{O_2}^{refine}$ | $F_i^{C \rightarrow CO_2} = \frac{v_i^{C \rightarrow CO_2}}{v_{O_2}^{C \rightarrow CO_2}} F_{O_2}^{C \rightarrow CO_2}$ |
| $[Si] + \{O_2\} \rightarrow (SiO_2)$ | $F_{O_2}^{Si \rightarrow SiO_2} = x_{Si}^{steel} F_{O_2}^{refine}$ | $F_i^{Si \rightarrow SiO_2} = \frac{v_i^{Si \rightarrow SiO_2}}{v_{O_2}^{Si \rightarrow SiO_2}} F_{O_2}^{Si \rightarrow SiO_2}$ |
| $2[Cr] + \frac{3}{2}\{O_2\} \rightarrow (Cr_2O_3)$ | $F_{O_2}^{Cr \rightarrow Cr_2O_3} = x_{Cr}^{steel} F_{O_2}^{refine}$ | $F_i^{Cr \rightarrow Cr_2O_3} = \frac{v_i^{Cr \rightarrow Cr_2O_3}}{v_{O_2}^{Cr \rightarrow Cr_2O_3}} F_{O_2}^{Cr \rightarrow Cr_2O_3}$ |
| $2[Al] + \frac{3}{2}\{O_2\} \rightarrow (Al_2O_3)$ | $F_{O_2}^{Al \rightarrow Al_2O_3} = x_{Al}^{steel} F_{O_2}^{refine}$ | $F_i^{Al \rightarrow Al_2O_3} = \frac{v_i^{Al \rightarrow Al_2O_3}}{v_{O_2}^{Al \rightarrow Al_2O_3}} F_{O_2}^{Al \rightarrow Al_2O_3}$ |

The equilibrium reaction constants $k_{eq}$ and dependence on species $i$ mass fractions $x_i^{steel}$ and $x_i^{slag}$ are adapted from Turkdogan [15] and are given in Tables A2–A4. The rates of change of different species are given in Table A5 and hold for both the inner and outer control volumes, where the steel mass fractions and temperatures can vary.

The reference states for all species and the equilibrium constants in Table A2 are the standard state of 25 °C and 1 atm.

**Table A2.** Equilibrium constants for reactions. $T_l$ denotes liquid phase temperatures.

| Reaction | Equilibrium Constant |
|---|---|
| $(FeO) + [C] \leftrightarrow Fe + \{CO\}$ | $\log_{10}\left(K_{eq}^{FeO \leftrightarrow CO}\right) = \frac{-5730}{\frac{1}{2}\left(T_l^{steel} + T_l^{slag}\right) + 273.15} + 5.096$ |
| $(FeO) + [Mn] \leftrightarrow Fe + (MnO)$ | $\log_{10}\left(K_{eq}^{FeO \leftrightarrow MnO}\right) = 2$ |
| $(MnO) + [C] \leftrightarrow [Mn] + \{CO\}$ | $\log_{10}\left(K_{eq}^{MnO \leftrightarrow CO}\right) = \frac{-13182}{\frac{1}{2}\left(T_l^{steel} + T_l^{slag}\right) + 273.15} + 8.574$ |
| $2(FeO) + [Si] \leftrightarrow 2Fe + (SiO_2)$ | $\log_{10}\left(K_{eq}^{FeO \leftrightarrow SiO_2}\right) = \frac{1510}{\frac{1}{2}\left(T_l^{steel} + T_l^{slag}\right) + 273.15} + 1.72$ |
| $2(MnO) + [Si] \leftrightarrow 2[Mn] + (SiO_2)$ | $\log_{10}\left(K_{eq}^{MnO \leftrightarrow SiO_2}\right) = \frac{1510}{\frac{1}{2}\left(T_l^{steel} + T_l^{slag}\right) + 273.15} + 1.27$ |
| $3(FeO) + 2[Cr] \leftrightarrow 3Fe + (Cr_2O_3)$ | $\log_{10}\left(K_{eq}^{FeO \leftrightarrow Cr_2O_3}\right) = 0.3$ |
| $3(SiO_2) + 4[Al] \leftrightarrow 3[Si] + 2(Al_2O_3)$ | $\log_{10}\left(K_{eq}^{SiO_2 \leftrightarrow Al_2O_3}\right) = \frac{17065}{\frac{1}{2}\left(T_l^{steel} + T_l^{slag}\right) + 273.15} - 14.465$ |

**Table A3.** Forward reaction rates for equilibrium reactions. $x$ denotes mass fractions and $k_f$ refers to kinetic model constants for each reaction.

| Reaction | Forward Reaction Rate | Units |
|---|---|---|
| $(FeO) + [C] \leftrightarrow Fe + (CO)$ | $F_f^{FeO \leftrightarrow CO} = (0.11 \cdot 10^4) K_f^{FeO \leftrightarrow CO} x_{FeO}^{slag} x_C^{steel}$ | $\frac{mol\ FeO}{s}$ |
| $(FeO) + [Mn] \leftrightarrow Fe + (MnO)$ | $F_f^{FeO \leftrightarrow MnO} = 10^4 \cdot K_f^{FeO \leftrightarrow MnO} x_{FeO}^{slag} x_{Mn}^{steel}$ | $\frac{mol\ FeO}{s}$ |
| $(MnO) + [C] \leftrightarrow [Mn] + \{CO\}$ | $F_f^{MnO \leftrightarrow CO} = (0.017 \cdot 10^4) K_f^{MnO \leftrightarrow CO} x_{MnO}^{slag} x_C^{steel}$ | $\frac{mol\ MnO}{s}$ |
| $2(FeO) + [Si] \leftrightarrow 2Fe + (SiO_2)$ | $F_f^{FeO \leftrightarrow SiO_2} = (2 \cdot 10^4) K_f^{FeO \leftrightarrow SiO_2} \left( x_{FeO}^{slag} \right)^2 x_{Si}^{steel}$ | $\frac{mol\ FeO}{s}$ |
| $2(MnO) + [Si] \leftrightarrow 2[Mn] + (SiO_2)$ | $F_f^{MnO \leftrightarrow SiO_2} = (2 \cdot 10^4) K_f^{MnO \leftrightarrow SiO_2} \left( x_{MnO}^{slag} \right)^2 x_{Si}^{steel}$ | $\frac{mol\ MnO}{s}$ |
| $3(FeO) + 2[Cr] \leftrightarrow 3Fe + (Cr_2O_3)$ | $F_f^{FeO \leftrightarrow Cr_2O_3} = (3 \cdot 10^4) K_f^{FeO \leftrightarrow Cr_2O_3} x_{FeO}^{slag} x_{Cr}^{steel}$ | $\frac{mol\ FeO}{s}$ |
| $3(SiO_2) + 4[Al] \leftrightarrow 3[Si] + 2(Al_2O_3)$ | $F_f^{SiO_2 \leftrightarrow Al_2O_3} = (1.5 \cdot 10^4) K_f^{SiO_2 \leftrightarrow Al_2O_3} x_{SiO_2}^{slag} x_{Al}^{steel}$ | $\frac{mol\ SiO_2}{s}$ |

**Table A4.** Backward reaction rates for equilibrium reactions. $x$ denotes mass fractions, $k_f$ refers to kinetic model constants for each reaction, $p_{CO}$ is the partial pressure of CO and $M$ denotes molar masses.

| Reaction | Backward Reaction Rate | Units |
|---|---|---|
| $(FeO) + [C] \leftrightarrow Fe + \{CO\}$ | $F_b^{FeO \leftrightarrow CO} = \frac{K_f^{FeO \leftrightarrow CO}}{K_{eq}^{FeO \leftrightarrow CO}} p_{CO}$ | $\frac{mol\ Fe\ (FeO)}{s}$ |
| $(FeO) + [Mn] \leftrightarrow Fe + (MnO)$ | $F_b^{FeO \leftrightarrow MnO} = 10^2 \cdot \frac{K_f^{FeO \leftrightarrow MnO}}{K_{eq}^{FeO \leftrightarrow MnO}} x_{MnO}^{slag}$ | $\frac{mol\ Fe\ (FeO)}{s}$ |
| $(MnO) + [C] \leftrightarrow [Mn] + \{CO\}$ | $F_b^{MnO \leftrightarrow CO} = 10^2 \cdot \frac{K_f^{MnO \leftrightarrow CO}}{K_{eq}^{MnO \leftrightarrow CO}} p_{CO} x_{Mn}^{steel}$ | $\frac{mol\ Mn\ (MnO)}{s}$ |
| $2(FeO) + [Si] \leftrightarrow 2Fe + (SiO_2)$ | $F_b^{FeO \leftrightarrow SiO_2} = (2 \cdot 10^2) \frac{K_f^{FeO \leftrightarrow SiO_2}}{K_{eq}^{FeO \leftrightarrow SiO_2}} x_{SiO_2}^{slag}$ | $\frac{mol\ Fe\ (FeO)}{s}$ |
| $2(MnO) + [Si] \leftrightarrow 2[Mn] + (SiO_2)$ | $F_b^{MnO \leftrightarrow SiO_2} = (2 \cdot 10^4) \frac{K_f^{MnO \leftrightarrow SiO_2}}{K_{eq}^{MnO \leftrightarrow SiO_2}} \left( x_{Mn}^{steel} \right)^2 x_{SiO_2}^{slag}$ | $\frac{mol\ Mn\ (MnO)}{s}$ |
| $3(FeO) + 2[Cr] \leftrightarrow 3Fe + (Cr_2O_3)$ | $F_b^{FeO \leftrightarrow SiO_2} = \left( \frac{6 \cdot 10^2 \cdot M_{Cr}}{M_{Cr_2O_3}} \right) \frac{K_f^{FeO \leftrightarrow Cr_2O_3}}{K_{eq}^{FeO \leftrightarrow Cr_2O_3}} x_{Cr_2O_3}^{slag}$ | $\frac{mol\ Fe\ (FeO)}{s}$ |
| $3(SiO_2) + 4[Al] \leftrightarrow 3[Si] + 2(Al_2O_3)$ | $F_b^{SiO_2 \leftrightarrow Al_2O_3} = (1.5 \cdot 10^4) \frac{K_f^{SiO_2 \leftrightarrow Al_2O_3}}{K_{eq}^{SiO_2 \leftrightarrow Al_2O_3}} x_{Si}^{steel} x_{Al_2O_3}^{slag}$ | $\frac{mol\ Si\ (SiO_2)}{s}$ |

**Table A5.** Species rates of change for each steel–slag equilibrium reaction.

| Reaction | Species $i$ Rate of Change |
|---|---|
| $(FeO) + [C] \leftrightarrow Fe + \{CO\}$ | $F_i^{FeO \leftrightarrow CO} = \frac{v_i^{FeO \leftrightarrow CO}}{v_{FeO}^{FeO \leftrightarrow CO}} \left( F_f^{FeO \leftrightarrow CO} - F_b^{FeO \leftrightarrow CO} \right)$ |
| $(FeO) + [Mn] \leftrightarrow Fe + (MnO)$ | $F_i^{FeO \leftrightarrow MnO} = \frac{v_i^{FeO \leftrightarrow MnO}}{v_{FeO}^{FeO \leftrightarrow MnO}} \left( F_f^{FeO \leftrightarrow MnO} - F_b^{FeO \leftrightarrow MnO} \right)$ |
| $(MnO) + [C] \leftrightarrow [Mn] + \{CO\}$ | $F_i^{MnO \leftrightarrow CO} = \frac{v_i^{MnO \leftrightarrow CO}}{v_{MnO}^{MnO \leftrightarrow CO}} \left( F_f^{MnO \leftrightarrow CO} - F_b^{MnO \leftrightarrow CO} \right)$ |
| $2(FeO) + [Si] \leftrightarrow 2Fe + (SiO_2)$ | $F_i^{FeO \leftrightarrow SiO_2} = \frac{v_i^{FeO \leftrightarrow SiO_2}}{v_{FeO}^{FeO \leftrightarrow SiO_2}} \left( F_f^{FeO \leftrightarrow SiO_2} - F_b^{FeO \leftrightarrow SiO_2} \right)$ |
| $2(MnO) + [Si] \leftrightarrow 2[Mn] + (SiO_2)$ | $F_i^{MnO \leftrightarrow SiO_2} = \frac{v_i^{MnO \leftrightarrow SiO_2}}{v_{MnO}^{MnO \leftrightarrow SiO_2}} \left( F_f^{MnO \leftrightarrow SiO_2} - F_b^{MnO \leftrightarrow SiO_2} \right)$ |
| $3(FeO) + 2[Cr] \leftrightarrow 3Fe + (Cr_2O_3)$ | $F_i^{FeO \leftrightarrow Cr_2O_3} = \frac{v_i^{FeO \leftrightarrow Cr_2}}{v_{FeO}^{FeO \leftrightarrow Cr_2O_3}} \left( F_f^{FeO \leftrightarrow Cr_2O_3} - F_b^{FeO \leftrightarrow Cr_2O_3} \right)$ |

The molar rates of change due to reactions $F_i \left( \frac{mol}{s} \right)$ can be combined with the molar mass of each component $M_i \left( \frac{g}{mol} \right)$ to calculate the total mass rates of change of all dynamic state components $W_i \left( \frac{kg}{s} \right)$:

$$W_i = \left(1.0 \times 10^{-3}\right) M_i \left( \sum_{\text{oxygen}} F_i^{\text{oxygen}} + \sum_{\text{equilibrium}} F_i^{\text{equilibrium}} \right) \tag{A14}$$

*Appendix A.3. Overall Heat and Mass Balances*

Many of the heat and mass balances refer to the area fractions $x_{\text{inner}}$ and $x_{\text{outer}}$:

$$x_{\text{inner}} = \left( \frac{r_{\text{inner}}}{r_{\text{furnace}}} \right)^2 \tag{A15}$$

$$x_{\text{outer}} = 1 - x_{\text{inner}}$$

Depending on the sub- and super-scripts, $x_{(\text{solid/liquid})}^{(\text{inner/outer})}$ refers to the solid (s) or liquid (l) mass fractions in the inner or outer control volume. Similarly, $x_{i,\,(\text{solid/liquid})}^{(\text{inner/outer})}$

In the following overall steel mass balances, $u_{\text{scrap}}$ refers to scrap metal input to the furnace.

$$\frac{\mathrm{d}m_{\text{solid}}^{\text{inner}}}{\mathrm{d}t} = x_{\text{inner}} u_{\text{scrap}} - r_{\text{melt}}^{\text{inner}} + r_{\text{freeze}}^{\text{inner}}$$

$$\frac{\mathrm{d}m_{\text{solid}}^{\text{outer}}}{\mathrm{d}t} = x_{\text{outer}} u_{\text{scrap}} - r_{\text{melt}}^{\text{outer}} + r_{\text{freeze}}^{\text{inner}}$$

$$\frac{\mathrm{d}m_{\text{liquid}}^{\text{inner}}}{\mathrm{d}t} = r_{\text{melt}}^{\text{inner}} - r_{\text{freeze}}^{\text{inner}} + \sum_{i,\,\text{steel}} W_{i,\,\text{steel}}^{\text{inner}} \tag{A16}$$

$$\frac{\mathrm{d}m_{\text{liquid}}^{\text{outer}}}{\mathrm{d}t} = r_{\text{melt}}^{\text{outer}} - r_{\text{freeze}}^{\text{outer}} + \sum_{i,\,\text{steel}} W_{i,\,\text{steel}}^{\text{outer}}$$

In the following overall slag mass balances, $u_{\text{slag}}$ refers to slag input to the furnace.

$$\frac{\mathrm{d}m_{\text{solid}}^{\text{slag}}}{\mathrm{d}t} = u_{\text{slag}} - r_{\text{melt}}^{\text{slag}} + r_{\text{freeze}}^{\text{slag}}$$

$$\frac{\mathrm{d}m_{\text{liquid}}^{\text{slag}}}{\mathrm{d}t} = r_{\text{melt}}^{\text{slag}} - r_{\text{freeze}}^{\text{slag}} + \sum_{i,\,\text{slag}} \left( W_{i,\,\text{slag}}^{\text{inner}} + W_{i,\,\text{slag}}^{\text{outer}} \right) \tag{A17}$$

The species *i* fraction of the charged scrap $x_i^{\text{scrap}}$ enters the scrap component balances:

$$\frac{\mathrm{d}m_{i,\,\text{solid}}^{\text{inner}}}{\mathrm{d}t} = x_{\text{inner}} u_{\text{scrap}} x_i^{\text{scrap}} - r_{\text{melt}}^{\text{inner}} x_{i,\,\text{solid}}^{\text{inner}} + r_{\text{freeze}}^{\text{inner}} x_{i,\,\text{liquid}}^{\text{inner}}$$

$$\frac{\mathrm{d}m_{i,\,\text{solid}}^{\text{outer}}}{\mathrm{d}t} = x_{\text{outer}} u_{\text{scrap}} x_i^{\text{scrap}} - r_{\text{melt}}^{\text{outer}} x_{i,\,\text{solid}}^{\text{outer}} + r_{\text{freeze}}^{\text{outer}} x_{i,\,\text{liquid}}^{\text{outer}}$$

$$\frac{\mathrm{d}m_{i,\,\text{liquid}}^{\text{inner}}}{\mathrm{d}t} = r_{\text{melt}}^{\text{inner}} x_{i,\,\text{solid}}^{\text{inner}} - r_{\text{freeze}}^{\text{inner}} x_{i,\,\text{liquid}}^{\text{inner}} + \sum_{i,\,\text{steel}} W_{i,\,\text{steel}}^{\text{inner}} \tag{A18}$$

$$\frac{\mathrm{d}m_{i,\,\text{liquid}}^{\text{outer}}}{\mathrm{d}t} = r_{\text{melt}}^{\text{outer}} x_{i,\,\text{solid}}^{\text{outer}} - r_{\text{freeze}}^{\text{outer}} x_{i,\,\text{liquid}}^{\text{outer}} + \sum_{i,\,\text{steel}} W_{i,\,\text{steel}}^{\text{outer}}$$

The species *i* fraction of the charged slag $x_i^{\text{slag}}$ enters the slag component balances:

$$\frac{\mathrm{d}m_{i,\,\text{solid}}^{\text{slag}}}{\mathrm{d}t} = u_{\text{slag}} x_i^{\text{slag}} - r_{\text{melt}}^{\text{slag}} x_{i,\,\text{solid}}^{\text{slag}} + r_{\text{freeze}}^{\text{slag}} x_{i,\,\text{liquid}}^{\text{slag}}$$

$$\frac{\mathrm{d}m_{i,\,\text{liquid}}^{\text{slag}}}{\mathrm{d}t} = r_{\text{melt}}^{\text{slag}} x_{i,\,\text{solid}}^{\text{slag}} - r_{\text{freeze}}^{\text{slag}} x_{i,\,\text{liquid}}^{\text{slag}} + \sum_{i_{\text{slag}}} W_{i_{\text{slag}}} \tag{A19}$$

The mol balance of CO gas is given:

$$\frac{\mathrm{d}n_{\mathrm{CO}}}{\mathrm{d}t} = \sum_{\text{equilibrium}} F_{\mathrm{CO}}^{\text{equilibrium}} - 2F_{\mathrm{O}_2}^{\mathrm{CO}} \tag{A20}$$

The temperature balances of steel phases:

$$\frac{\mathrm{d}T_{\text{solid}}^{\text{inner}}}{\mathrm{d}t} = \frac{\begin{aligned}&x_{\text{solid}}^{\text{inner}} P_{\text{inner}} - Q_{\text{solid–liquid}}^{\text{inner-inner}} - Q_{\text{solid–solid}}^{\text{inner-outer}} - Q_{\text{solid–liquid}}^{\text{inner-outer}} - Q_{\text{solid-roof}}^{\text{inner}} - Q_{\text{solid-vessel}}^{\text{inner}} - Q_{\text{solid-gas}}^{\text{inner}}\\&\quad - Q_{\text{solid–solid}}^{\text{inner-slag}} - Q_{\text{solid–liquid}}^{\text{inner-slag}} + Q_{\text{freeze}}^{\text{inner}} - Q_{\text{melt}}^{\text{inner}} - C_{\text{p,solid}} x_{\text{inner}} u_{\text{scrap}} \left( T_{\text{solid}}^{\text{inner}} - T_{\text{ambient}} \right)\end{aligned}}{C_{\text{p, solid}} m_{\text{solid}}^{\text{inner}}}$$

$$\frac{\mathrm{d}T_{\text{solid}}^{\text{outer}}}{\mathrm{d}t} = \frac{\begin{aligned}&x_{\text{solid}}^{\text{outer}} P_{\text{outer}} - Q_{\text{solid–liquid}}^{\text{outer-outer}} + Q_{\text{solid–solid}}^{\text{inner-outer}} - Q_{\text{liquid–solid}}^{\text{inner-outer}} - Q_{\text{solid-roof}}^{\text{outer}} - Q_{\text{solid-vessel}}^{\text{outer}} - Q_{\text{solid-gas}}^{\text{outer}}\\&\quad - Q_{\text{solid–solid}}^{\text{outer-slag}} - Q_{\text{solid–liquid}}^{\text{outer-slag}} + Q_{\text{freeze}}^{\text{outer}} - Q_{\text{melt}}^{\text{outer}} - C_{\text{p,solid}} x_{\text{outer}} u_{\text{scrap}} \left( T_{\text{solid}}^{\text{outer}} - T_{\text{ambient}} \right)\end{aligned}}{C_{\text{p, solid}} m_{\text{solid}}^{\text{outer}}}$$

$$\frac{\mathrm{d}T_{\text{liquid}}^{\text{inner}}}{\mathrm{d}t} = \frac{\begin{aligned}&x_{\text{liquid}}^{\text{inner}} P_{\text{inner}} + Q_{\text{solid–liquid}}^{\text{inner-inner}} - Q_{\text{liquid–solid}}^{\text{inner-outer}} - Q_{\text{liquid–liquid}}^{\text{inner-outer}} - Q_{\text{liquid-roof}}^{\text{inner}} - Q_{\text{liquid-vessel}}^{\text{inner}} - Q_{\text{liquid-gas}}^{\text{inner}}\\&\quad - Q_{\text{liquid–solid}}^{\text{inner-slag}} - Q_{\text{liquid–liquid}}^{\text{inner-slag}} + Q_{\text{freeze}}^{\text{inner}} - Q_{\text{melt}}^{\text{inner}}\end{aligned}}{C_{\text{p, liquid}} m_{\text{liquid}}^{\text{inner}}}$$

$$\frac{\mathrm{d}T_{\text{liquid}}^{\text{outer}}}{\mathrm{d}t} = \frac{\begin{aligned}&x_{\text{liquid}}^{\text{outer}} P_{\text{outer}} + Q_{\text{solid–liquid}}^{\text{outer-outer}} + Q_{\text{solid–liquid}}^{\text{inner-outer}} + Q_{\text{liquid–liquid}}^{\text{inner-outer}} - Q_{\text{liquid-roof}}^{\text{outer}} - Q_{\text{liquid-vessel}}^{\text{outer}} - Q_{\text{liquid-gas}}^{\text{outer}}\\&\quad - Q_{\text{liquid–solid}}^{\text{outer-slag}} - Q_{\text{liquid–liquid}}^{\text{outer-slag}} + Q_{\text{freeze}}^{\text{outer}} - Q_{\text{melt}}^{\text{outer}}\end{aligned}}{C_{\text{p, liquid}} m_{\text{liquid}}^{\text{outer}}} \tag{A21}$$

The temperature balances of slag phases:

$$\frac{\mathrm{d}T_{\text{solid}}^{\text{slag}}}{\mathrm{d}t} = \frac{\begin{aligned}&Q_{\text{solid–solid}}^{\text{inner-slag}} + Q_{\text{solid–solid}}^{\text{outer-slag}} + Q_{\text{liquid–solid}}^{\text{inner-slag}} + Q_{\text{liquid–solid}}^{\text{outer-slag}}\\&\quad + Q_{\text{freeze}}^{\text{slag}} - Q_{\text{melt}}^{\text{slag}} - C_{\text{p, slag}} u_{\text{slag}} \left( T_{\text{solid}}^{\text{slag}} - T_{\text{ambient}} \right)\end{aligned}}{C_{\text{p, slag}} m_{\text{solid}}^{\text{slag}}}$$

$$\frac{\mathrm{d}T_{\text{liquid}}^{\text{slag}}}{\mathrm{d}t} = \frac{\begin{aligned}&Q_{\text{solid–liquid}}^{\text{inner-slag}} + Q_{\text{solid–liquid}}^{\text{outer-slag}} + Q_{\text{liquid–liquid}}^{\text{inner-slag}} + Q_{\text{liquid–liquid}}^{\text{outer-slag}}\\&\quad\quad\quad\quad\quad\quad\quad\quad\quad\quad + Q_{\text{freeze}}^{\text{slag}} - Q_{\text{melt}}^{\text{slag}}\end{aligned}}{C_{\text{p, slag}} m_{\text{liquid}}^{\text{slag}}} \tag{A22}$$

The temperature balances of gas phases:

$$\frac{\mathrm{d}T_{\text{gas}}}{\mathrm{d}t} = \frac{\begin{aligned}&P_{\text{gas}} + Q_{\text{solid-gas}}^{\text{inner}} + Q_{\text{solid-gas}}^{\text{outer}} + Q_{\text{liquid-gas}}^{\text{inner}} + Q_{\text{liquid-gas}}^{\text{outer}} - Q_{\text{gas-roof}} - Q_{\text{gas-vessel}}\\&\quad - F_{\text{LNG}} C_{\text{p, LNG}} \left( T_{\text{gas}} - T_{\text{ambient}} \right) - F_{\mathrm{O}_2} C_{\text{p, O}_2} \left( T_{\text{gas}} - T_{\text{ambient}} \right)\end{aligned}}{C_{\text{p, gas}} n_{\text{gas}}}$$

$$\frac{\mathrm{d}T_{\text{offgas}}}{\mathrm{d}t} = \frac{F_{\text{offgas}} C_{\text{p, gas}} \left( T_{\text{gas}} - T_{\text{offgas}} \right) + 0.05 F_{\text{air}} C_{\text{p, air}} \left( T_{\text{ambient}} - T_{\text{offgas}} \right)}{2 C_{\text{p, gas}} n_{\text{gas}}} \tag{A23}$$

$n_{\text{gas}}$ is calculated from the ideal gas law using standard temperature and pressure (STP) conditions ($T = 0\,°\mathrm{C}$, $p = 100$ kPa) and the furnace volume given by the dimensions in Appendix A.4. $C_{\text{p, gas}}$ is calculated from the LNG and oxygen flow rates and heat capacities:

$$C_{\text{p, gas}} = \frac{F_{\text{LNG}} C_{\text{p, LNG}} + F_{\mathrm{O}_2} C_{\text{p, O}_2}}{F_{\text{LNG}} + F_{\mathrm{O}_2}} \tag{A24}$$

The temperature balances of furnace equipment:

$$\frac{dT_{\text{roof}}}{dt} = \frac{\begin{array}{c} P_{\text{arc}}^{\text{roof}} + Q_{\text{solid-roof}}^{\text{inner}} + Q_{\text{solid-roof}}^{\text{outer}} + Q_{\text{liquid-roof}}^{\text{inner}} + Q_{\text{liquid-roof}}^{\text{outer}} \\ + Q_{\text{gas-roof}} - Q_{\text{roof-water}} - Q_{\text{roof-vessel}} \end{array}}{C_{\text{p, roof}} m_{\text{roof}}}$$

$$\frac{dT_{\text{vessel}}}{dt} = \frac{\begin{array}{c} P_{\text{arc}}^{\text{vessel}} + Q_{\text{solid-vessel}}^{\text{inner}} + Q_{\text{solid-vessel}}^{\text{outer}} + Q_{\text{liquid-vessel}}^{\text{inner}} + Q_{\text{liquid-vessel}}^{\text{outer}} \\ + Q_{\text{gas-vessel}} - Q_{\text{vessel-water}} + Q_{\text{roof-vessel}} \end{array}}{C_{\text{p, vessel}} m_{\text{vessel}}} \tag{A25}$$

## Appendix A.4. Model Constants and Dimensions

**Table A6.** Model constants and dimensions. An empty entry in the **Units** column indicates a unitless quantity.

| Constant | Description | Value | Units |
|---|---|---|---|
| $d_{\text{furnace}}$ | Furnace diameter | 8.1 | m |
| $d_{\text{inner}}$ | Inner control volume diameter | 4.65 | m |
| $h_{\text{furnace}}$ | Furnace height | 5.2 | m |
| $h_{\text{panel}}$ | Cooling water panel height | 2.89 | m |
| $C_{\text{p, solid}}$ | Heat capacity of solid steel | 39 | $\frac{\text{J}}{\text{mol K}}$ |
| $C_{\text{p, liquid}}$ | Heat capacity of liquid steel | 46 | $\frac{\text{J}}{\text{mol K}}$ |
| $C_{\text{p, slag}}$ | Heat capacity of slag | 50 | $\frac{\text{J}}{\text{mol K}}$ |
| $\rho_{\text{solid}}$ | Density of solid steel | 2000 | $\frac{\text{kg}}{\text{m}^3}$ |
| $\rho_{\text{liquid}}$ | Density of liquid steel | 7000 | $\frac{\text{kg}}{\text{m}^3}$ |
| $k_{\text{phase}}$ | Phase change constant | 0.005 | $\frac{1}{\text{s}}$ |
| $k_{\text{CO}}$ | Limiting constant for CO combustion | 0.25 | $\frac{1}{\text{s}}$ |
| $x_{\text{arc}}^{\text{gas}}$ | Fraction of arc power used to heat gas | 0.05 | |
| $x_{\text{vessel}}^{\text{loss}}$ | Fraction of arc losses used to heat vessel | 0.3 | |
| $k_{\text{ss}}$ | Heat transfer coefficient: solid steel–solid steel | 400 | $\frac{\text{W}}{\text{m}^2\text{K}}$ |
| $k_{\text{sl}}$ | Heat transfer coefficient: solid steel–liquid steel | 12,000 | $\frac{\text{W}}{\text{m}^2\text{K}}$ |
| $k_{\text{ll}}$ | Heat transfer coefficient: liquid steel–liquid steel | 60,000 | $\frac{\text{W}}{\text{m}^2\text{K}}$ |
| $k_{\text{cs}}$ | Heat transfer coefficient: solid slag–solid steel | 2000 | $\frac{\text{W}}{\text{m}^2\text{K}}$ |
| $k_{\text{cl}}$ | Heat transfer coefficient: solid slag–liquid steel | 2000 | $\frac{\text{W}}{\text{m}^2\text{K}}$ |
| $k_{\text{bs}}$ | Heat transfer coefficient: liquid slag–solid steel | 5 | $\frac{\text{W}}{\text{m}^2\text{K}}$ |
| $k_{\text{bl}}$ | Heat transfer coefficient: liquid slag–liquid steel | 5 | $\frac{\text{W}}{\text{m}^2\text{K}}$ |
| $k_{\text{sg}}$ | Heat transfer coefficient: solid steel–gas | 20 | $\frac{\text{W}}{\text{m}^2\text{K}}$ |
| $k_{\text{lg}}$ | Heat transfer coefficient: liquid steel–gas | 10 | $\frac{\text{W}}{\text{m}^2\text{K}}$ |
| $k_{\text{gr}}$ | Heat transfer coefficient: gas–roof | 25 | $\frac{\text{W}}{\text{m}^2\text{K}}$ |
| $k_{\text{gv}}$ | Heat transfer coefficient: gas–vessel | 25 | $\frac{\text{W}}{\text{m}^2\text{K}}$ |
| $k_{\text{rw}}$ | Heat transfer coefficient: roof–water | 300 | $\frac{\text{W}}{\text{m}^2\text{K}}$ |
| $k_{\text{vw}}$ | Heat transfer coefficient: vessel–water | 300 | $\frac{\text{W}}{\text{m}^2\text{K}}$ |
| $\varepsilon_{\text{s}}$ | Emissivity of solid steel | 0.4 | |
| $\varepsilon_{\text{l}}$ | Emissivity of liquid steel | 0.6 | |
| $\varepsilon_{\text{r}}$ | Emissivity of furnace roof | 0.7 | |
| $\varepsilon_{\text{v}}$ | Emissivity of side panels | 0.5 | |

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
