# Peer review of "Thermophysical Model for Online Optimization and Control of the Electric Arc Furnace"

_metals, doi:10.3390/met11101587_

Round 1
Reviewer 1 Report
Dear Authors,
Your paper "Thermophysical Model for Online Optimization and Control of the Electric Arc Furnace”" develop a dynamic control model in EAF steelmaking process, which aims to optimize the electric power profile and electric arc operation. It has the potential to be a nice paper assuming that the following corrections are made:
- In introduction, some of the mentioned MPC algorithm employed are linear, while others are non-linear. However, there is a lack of comparing description of their advantages and disadvantages. Why do you choose the non-linear scheme in your model. Whether it will bring additional benefits?
- Figure 1, what is the basis for dividing the bath area into red and yellow parts? Please define a standard in the article and give the scientific basis.
- Table 1, why the Mn do not reactive with O2?
- Line 59, please indicate the meaning of the acronym “BFI”
- It should be noted that the shortcomings and weaknesses of relevant models listed in introduction were not performed. It is not possible to get some advanced features of the model described in this paper. So some relative contents are suggested in introduction.
- Line 118~119, Heat transfer between steel and the slag masses is then tuned to indirectly account for the interactions of slag with the environment?
- Lines 84-86, The MPC application of the proposed model uses a finite receding horizon, where the duration of the heat is not specified but the predictions horizon becomes shorter as the process nears completion. What do the words mean.
- The BFI model was not mentioned in the model of this paper. Why did the author describe it in introduction.
- All descriptions of the figures are recommended to be placed in the content instead of listed separately after the name of the figure. Please modify them.
- In figure 1, The side panels and roof exchange heat with cooling water and are marked on the diagram with arrows. As we know, the direction of heat transfer should be from the hot object to the cold object. So the application of two-way arrows is not appropriate.Besides, one side of the arrow should not point to the gas in the furnace. Please correct them.
- In the description of Figure 1, “slag state variables do not exchange heat directly with the furnace environment?” As we know, the slag surface is still subjected to the high temperature radiant heat from the furnace wall and electrode and the convection heat transfer between the gas in the furnace and the slag surface.
- In figure 1, state variables can not be identified by bold text. Please try to distinguish them by font size or color.
- In table 2, the liquid and solid phases inside the molten bath are not in direct contact with the gas, so there is no convection or heat conduction. The same problems also exsit in Table 3. Please explain them.
- In equation 4, the area for heat transfer is well defined and analysed, while I can not see the related content of heat transfer coefficient. How do you address it in your paper.
- In Figure 2, please show the reason why the solid temperature (blue line) increases in a straight line while liquid temperature (red line) decreased in a curve during the melting process.
- In the abstract, the MPC applications derived from this process model can result in electrical energy consumption savings of 1–2%. But I can not find the relevant data in your article text. How do you get the propertion of energy saving of 1-2%.
Reviewer 2 Report
The paper is interesting and well written. The purpose of the paper is underlined. Conclusions are a bit too short, but valuable. Methodology is clear and understandable, the same the results.
However, I have small problems:
- Problems in presentation of Figures and Tables. For me, titles of Figures and Tables are too long. You should shorten them (Figures 1-10). Some parts of the texts are explanations of Figures. These parts can be moved to normal text of the paper. The same with Tables 2-4. Maybe you can prepare just short legend under the Figures and Tables. Please, think about it. It will be more understandable for readers.
- Figure 4 – lack of description of the horizontal axis. You use words “top” and “bottom”. Maybe “a” and “b” is better.
- Figure 6, 7, 9, 10 – “a” and “b” instead of “top and “bottom” (or other words)
Reviewer 3 Report
More justification is needed of the assumption that the slag does not exchange heat with the furnace freeboard or (apparently) with the arc. This seems unreasonable for furnaces that practice slag foaming.
The way in which heat transfer from the arc is treated must be stated more clearly. It is stated in section 3.1 that "arc power is directly supplied to the inner steel control volume", but this should be stated (and justified) in the model description (with reference to the relevant section in the Appendix).
Clarify why endpoint temperature was not included as an optimization target (line 363).
Please use the capital K for equilibrium constants.
State the reference states for the equilibrium constants given in the Appendix.
Given the expected increase in the importance of direct-reduced iron (DRI) as feed, please comment on the applicability of the model to DRI-based operations.
Reviewer 4 Report
See attached comments.

Reviewer 5 Report
The paper presents a dynamic, first-principles thermo-physical model for a steelmaking Electric Arc Furnace (EAF), which is integrated into an application designed for EAF process optimization during operations.
The topic of the paper is very interesting, of practical industrial relevance and perfectly fits the aims and scope of the Journal.
The paper is overall well written and nice to read.
The introduction provides a complete and detailed analysis of the state of the art encompassing the main relevant works in the field of mathematical modelling for forecasting the behavior of the EAF and the application of Model Predictive Control (MPC) for the EAF process.
The main relevant element of novelty of the proposed model, as presented by the authors, is the inclusion of auxiliary process data to improve forecasting of energy efficiency and heat transfer limitations in the EAF. In particular, a more detailed modelling is provided concerning different control volumes with liquid and solid phases to estimate the visibility of the electric arc and arc efficiency for melting and heating. A Non-linear Model Predictive Control framework is applied for real-time optimization, which exploits a finite receding horizon, with unspecified duration of the heat but with forecasting horizon shortening as the process nears completion. The novelty of the proposed approach is consistent and well described and duly highlighted at the end of the introductory section.
The pursued modelling approach is technically sound and comprehensive. The developed models are generally well detailed in Section 2. However, the Solid-Liquid Phase Change (Subsection 2.4) is poorly described and Figure 2 is not particularly supportive. Moreover, the axes are not indicated and the caption is not fully clear. This part must be improved.
The results are significant, interesting and comprehensively described and discussed in Section 3. However, some improvements are needed. Figure 4, that describes the outcome of the model of the meltdown behavior should be more deeply commented in the text and lacks of the indication of the horizontal axis. Also Figure 7 lacks of indication of the horizontal axis. Finally, the exemplar description of the MPC-based optimization of batch time and electric arc efficiency, which is supported by Figure 8, in not really effective and should be more deeply explained.
The conclusions are clear and overall supported by the presented results.
As a minor formal remark, the use of acronyms should be avoided in the abstract.
Round 2
Reviewer 4 Report
Thank you for your response to my comments.
Reviewer 5 Report
The Authors carefully revised the paper according to the observations and suggestions provided by the reviewers. As a consequence, all the major weaknesses of the paper have been overcome and the quality of the paper is now suitable to publication.